# DISCRETE DIFFUSION LANGUAGE MODELING BY ESTIMATING THE RATIOS OF THE DATA DISTRIBUTION

## ABSTRACT

Despite their groundbreaking performance for many generative modeling tasks, diffusion models have fallen short on discrete data domains such as natural language. Crucially, standard diffusion models rely on the well-established theory of score matching, but efforts to generalize this to discrete structures have not yielded the same empirical gains. In this work, we bridge this gap by proposing score entropy, a novel discrete score matching loss that is more stable than existing methods, forms an ELBO for maximum likelihood training, and can be efficiently optimized with a denoising variant. We scale our Score Entropy Discrete Diffusion models (SEDD) to the experimental setting of GPT-2, achieving highly competitive likelihoods while also introducing distinct algorithmic advantages. In particular, when comparing similarly sized SEDD and GPT-2 models, SEDD attains comparable perplexities (normally within $+10\%$ of and sometimes outperforming the baseline). Furthermore, SEDD models learn a more faithful sequence distribution (around $4\times$ better compared to GPT-2 models with ancestral sampling as measured by large models), can trade off compute for generation quality (needing $16\times$ fewer network evaluations to match GPT-2 quality), and enables arbitrary infilling beyond the standard left to right prompting.

## 1 INTRODUCTION

Many recent advances in deep learning have centered around generative modeling. In this setting, neural networks learn to generate new samples given unstructured data. Remarkably, combining the powerful generalization of neural networks with this rather straightforward objective has led to unparalleled capabilities. For example, modern "generative AI" systems are able to generate images from arcane descriptions (Ramesh et al., 2022) and answer complex queries (Brown et al., 2020).

So far, the techniques used for these advances have largely been bifurcated according to the structure of the data. For computer vision data, where one can faithfully dequantize into continuous space, diffusion modeling (Sohl-Dickstein et al., 2015; Ho et al., 2020; Song et al., 2021b) is the core paradigm undergirding contemporary methods. Conversely, for natural language data, which is far more discrete, autoregressive modeling is indispensable for most problems (Radford et al., 2019).

Despite this field-level divide, researchers have attempted to apply diffusion models to language modeling tasks (Li et al., 2022; Austin et al., 2021). This can be done by either designing a special dequantization for the discrete tokens (Dieleman et al., 2022) or by directly modeling a discrete diffusion process on said tokens (He et al., 2022; Zheng et al., 2023). However, despite considerable efforts, no such method has yet yielded a diffusion model scheme that is on par with or provides a clear benefit over standard autoregressive training.

In our work, we close this gap for GPT-2 scale experiments (Radford et al., 2019), demonstrating, for the first time, a non-autoregressive modeling technique that is able to achieve similar perplexity scores as autoregressive modeling at a large scale. Our approach has the added benefits of producing better samples from the learned distribution, allowing for a compute-quality tradeoff, and enabling prompting with arbitrary positions. Key to this success is score entropy, a novel training objective for discrete-space diffusion models that is analogous to score matching for continuous-space diffusion models (Hyvärinen, 2005; Song & Ermon, 2019). Our contributions can be summarized as follows:

1. We introduce score entropy, a discrete score matching loss that can be used to trained discrete diffusion models. Score entropy learns the concrete scores (analogous to the score function in standard diffusion) of the perturbed data distribution in a scalable and principled manner.

2. We use the modeled scores to develop several enhanced sampling methods. In particular, we derive a score-based ancestral sampling method and a general infilling procedure.

3. We combine our theoretical advances with architectural improvements to scale our Score Entropy Discrete Diffusion models (SEDD) to GPT-2 model sizes. As previously mentioned, SEDD is comparable to GPT-2 for perplexities but also offer several distinct advantages for high quality, fast, and controllable generation.

## 2 PRELIMINARIES

Diffusion models learn to generate data by reversing a Markov process that takes the data distribution $x_0 \sim p_0$ to a simple noise distribution $x_T \sim p_T$ (Sohl-Dickstein et al., 2015). We want to reverse this process, allowing us to sample from $p_T$ to generate samples from $p_0$, but the reverse transitions $p(x_{t-\Delta t}|x_t)$ are difficult to approximate since they are nontrivial densities. However, as $\Delta t \to 0$, the concept of the "score function" emerges to enable a more faithful modeling paradigm. Learning this quantity is well-established for continuous spaces but remains an open problem for discrete spaces.

### 2.1 CONTINUOUS DIFFUSION MODELS

When the data support is $\mathbb{R}^d$, one constructs the Markov process by perturbing data points $\mathbf{x}_0 \sim p_0$ with a stochastic process defined by the stochastic differential equation (SDE) (Song et al., 2021b):

$$d\mathbf{x}_t = f(\mathbf{x}_t, t)dt + g(t)d\mathbf{B}_t \tag{1}$$

The perturbed densities $p_t$ of the points $\mathbf{x}_t$ evolve according to the corresponding Fokker-Planck partial differential equation and approaches a Gaussian limit distribution $\pi \approx p_T$. A famous result by Anderson constructs the reverse of this stochastic differential equation (Anderson, 1982):

$$d\mathbf{x}_t = \left( f(\mathbf{x}_t, t) - g(t)^2 \frac{\nabla_x p_t(\mathbf{x}_t)}{p_t(\mathbf{x}_t)} \right) dt + g(t)d\mathbf{B}_t \tag{2}$$

which takes $p_T$ back to $p_0$. One approximates this process by learning the unknown $\frac{\nabla_x p_t(\mathbf{x}_t)}{p_t(\mathbf{x}_t)}$ (normally written as $\nabla_x \log p_t(\mathbf{x}_t)$) with a neural network $\mathbf{s}_\theta(\mathbf{x}_t, t)$. This can be done optimizing the well known score matching (shown below) jointly over all $t$.

$$\mathcal{L}_{\text{SM}} = \frac{1}{2} \mathbb{E}_{\mathbf{x} \sim p_t} \left\| \mathbf{s}_\theta(\mathbf{x}, t) - \frac{\nabla_x p_t(\mathbf{x})}{p_t(\mathbf{x})} \right\|^2 \tag{3}$$

Score matching has many equivalent forms such as the implicit (Song et al., 2019) and denoising score matching losses (Vincent, 2011) that remove the unknown term $\frac{\nabla_x p_t(\mathbf{x})}{p_t(\mathbf{x})}$. With a learned score, the diffusion model can sample $x_T \sim \pi$ and solves the parameterized reverse SDE

$$d\mathbf{x}_t = \left( f(\mathbf{x}_t, t) - g(t)^2 \mathbf{s}_\theta(\mathbf{x}, t) \right) dt + g(t)d\mathbf{B}_t \tag{4}$$

to approximately sample from $p_0$. Importantly, when the score matching losses are jointly optimized with a relative weighting of $g(t)^2$ for each $t$, one is able to compute the ELBO for training and evaluating likelihoods (Song et al., 2021a; Kingma et al., 2021; Huang et al., 2021).

This so-called "score-parameterization" of diffusion models has been essential for the recent success of continuous space diffusion models. In particular, modeling the score function (up to a scaling) has consistently been shown to result in both superior generation quality and improved likelihood values (Ho et al., 2020; Karras et al., 2022).

### 2.2 DISCRETE DIFFUSION MODELS

We now consider the a discrete data support $\{1, \dots, N\}$. Here, the probability distributions $p_t : \mathcal{X} \to R$ instead become probability mass vectors $p_t \in \mathbb{R}^N$ that are positive and sum to 1, and the diffusion

is best described by the discrete analogue of the Fokker-Planck partial differential equation acting on $p_t$. In particular, we evolve the data distribution $p_0$ according to the equation (Campbell et al., 2022)

$$\frac{dp_t}{dt} = Q_t p_t \quad p_0 = p_{\text{data}} \tag{5}$$

where $Q_t$ are the diffusion matrices that are required to have non-negative non-diagonal entries and columns which sum to zero (so that the rate $\frac{dp_t}{dt}$ sums to 0, meaning $p_t$ does not gain or lose total mass). This process is realized at the sample level by the transition densities that are defined by the columns of $Q_t$:

$$p(x_{t+\Delta t} = y | x_t = x) = \delta_{xy} + Q_t(y, x)\Delta t + O(\Delta t^2) \tag{6}$$

which enables a Euler-Maruyama type sampling algorithm that steps according to $\Delta t$. For certain $Q_t$, $p_T$ approaches a limiting distribution $\pi$ for large $T$. Additionally, this Markov process has a well known reversal (Kelly, 1980; Sun et al., 2023) given by another diffusion matrix $\overline{Q}_t$:

$$\frac{dp_{T-t}}{dt} = \overline{Q}_{T-t} p_{T-t} \quad \overline{Q}_t(y, x) = \begin{cases} \frac{p_t(y)}{p_t(x)} Q_t(x, y) & x \neq y \\ -\sum_{k \neq i} \overline{Q}_t(z, x) & x = y \end{cases} \tag{7}$$

Note that the reverse process again depends on $p_t$, which is defined by the data distribution $p_0$ and the diffusion $Q_t$. This is analogous to the reverse SDE (Equation 2), with the ratio $\frac{p_t(y)}{p_t(x)}$, generalizing the score function[1]. Similar to the continuous case, by approximating this score function, one can generate samples by sampling from $\pi$ and simulating a parameterized reverse process. However, there still has yet to be a consensus on how to learn these ratios (see Section 6).

**Concrete Score Matching.** Meng et al. (2022) take a score matching view and group $\left[\frac{p_t(y)}{p_t(x)}\right]_{y \neq x}$ for each value $x$, forming the *concrete score*. By generalizing the standard score matching loss, they learn $s_\theta(x, t) \approx \left[\frac{p_t(y)}{p_t(x)}\right]_{y \neq x}$ with a discrete generalization of the score matching loss:

$$\mathcal{L}_{\text{CSM}} = \frac{1}{2} \mathbb{E}_{x \sim p_t} \left[ \sum_{y \neq x} \left( s_\theta(x_t, t)_y - \frac{p_t(y)}{p_t(x)} \right)^2 \right] \tag{8}$$

Due to its similarities with standard score matching, this approach is rather promising. In particular, $s_\theta(x, t)$ is a general model and one recovers the true score given infinite data. However, in practice $\mathcal{L}_{\text{CSM}}$ is based on the $\ell^2$ loss, which is only suitable for real value inputs. Both $s_\theta(x, t)_j$ and $\frac{p(j)}{p(i)}$ are nonnegative, and this mismatch leads to suboptimal gradient behavior. As an example, $s_\theta(i, t)_j = 0$ and $0.2$ induce equal loss signals when $\frac{p(j)}{p(i)} = 0.1$, but the $0$ value is much worse: dropping the support of the data distribution induces an infinite KL divergence. As such, concrete score matching does not work for our large-scale diffusion modeling (see Appendix C.2 for ablation).

## 3 SCORE ENTROPY DISCRETE DIFFUSION MODELS

In this section, we introduce score entropy, our proposed loss. Similar to concrete score matching, we model the concrete score $s_\theta(x, t) \approx \left[\frac{p_t(y)}{p_t(x)}\right]_{y \neq x}$. However, we design this loss to be compatible with the modeled values and the discrete diffusion, necessitating a significantly different expression.

**Definition 3.1.** *We define the **score entropy** for a discrete distribution $p$, weights $w_{xy} \geq 0$ and a score network $s_\theta(x)_y$ as*

$$\mathcal{L}_{\text{SE}} = \mathbb{E}_{x \sim p} \left[ \sum_{y \neq x} w_{xy} \left( s_\theta(x)_y - \frac{p(y)}{p(x)} \log s_\theta(x)_y + K\left( \frac{p(y)}{p(x)} \right) \right) \right] \tag{9}$$

*where $K(a) = a(\log a - 1)$ is a normalizing constant function.*

---

[1]The gradient operator for discrete structures is (up to some scaling) defined for pairs $x \neq y$ by $\nabla f(xy) := f(y) - f(x)$. The score function would generalize to the normalized gradients $\frac{\nabla p(xy)}{p(x)} = \frac{p(y)}{p(x)} - 1$.

**Remark.** *Score entropy replaces the standard $\ell^2$ Euclidean distance metric with a Bregman divergence $D_F\left(s(x)_y, \frac{p(y)}{p(x)}\right)$ where $F = -\log$ is the convex function. As such, it is non-negative, reflexive, and convex. It generalizes the cross-entropy loss function to general positive values (as opposed to probabilities), inspiring the name. The weights $w_{xy}$ are similarity weights between $x$ and $y$ are used primarily when combining score entropy with diffusion models.*

While this expression is more complex than the standard score matching variants, we show that the score entropy satisfies several desiderata for a discrete diffusion training objective:

## 3.1 SCORE ENTROPY PROPERTIES

**First,** score entropy is a suitable loss function that recovers the ground truth concrete score.

**Proposition 3.2** (Consistency of Score Entropy). *Suppose $p$ is fully supported and $w_{xy} > 0$. As the number of samples and model capacity approaches $\infty$, the optimal $\theta^*$ that minimizes Equation 9 satisfies $s_{\theta^*}(x)_y = \frac{p(y)}{p(x)}$. Furthermore, $\mathcal{L}_{\text{SE}}$ will be 0 at $\theta^*$.*

**Second,** score entropy directly improves upon concrete score matching by rescaling problematic gradients. For the weights $w_{xy} = 1$, $\nabla_{s(x)_y}\mathcal{L}_{\text{SE}} = \frac{1}{s(x)_y}\nabla_{s(x)_y}\mathcal{L}_{\text{CSM}}$, so the gradient signals for each pair $(x, y)$ are scaled by a factor of $s(x)_y$ as a normalization component. As such, this forms a natural log-barrier which keeps our $s_\theta$ valid, as shown in Figure 1.

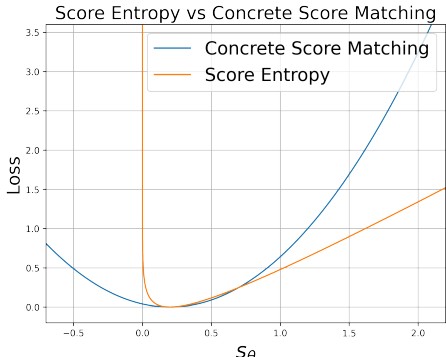

Figure 1: The graphs of $\mathcal{L}_{\text{CSM}}$ versus $\mathcal{L}_{\text{SE}}$ for a ground truth score of $0.2$. The score entropy loss respects nonnegativity.

**Third,** similar to concrete score matching, score entropy can be made computationally tractable by removing the unknown $\frac{p(y)}{p(x)}$. There are two alternative forms, the first of which is analogous to the implicit score matching loss (Hyvärinen, 2005):

**Proposition 3.3** (Implicit Score Entropy). *$\mathcal{L}_{\text{SE}}$ is equal up to a constant independent of $\theta$ to the **implicit score entropy***

$$\mathcal{L}_{\text{ISE}} = \mathbb{E}_{x \sim p}\left[\sum_{y \neq x} w_{xy} s_\theta(x)_y - w_{yx} \log s_\theta(y)_x\right] \quad (10)$$

We need to evaluate $s(y)$ for all $y$, which is intractable, so one must resort to sampling $y$ uniformly. This is analogous to the additional variance introduced by the Hutchinson trace estimator (Hutchinson, 1989) for sliced score matching (Song et al., 2019) and, in practice, renders $\mathcal{L}_{\text{ISE}}$ unsuitable for high dimensional problems. Therefore, we work with the score entropy variant of the more empirically practical denoising score matching loss (Vincent, 2011):

**Theorem 3.4** (Denoising Score Entropy). *Suppose $p$ is a perturbation of a base density $p_0$ and a transition kernel $p(\cdot|\cdot)$, ie $p(x) = \sum_{x_0} p(x|x_0)p_0(x_0)$. The score entropy is equivalent (up to a constant independent of $\theta$) to the **denoising score entropy***

$$\mathcal{L}_{\text{DSE}} = \mathbb{E}_{x_0 \sim p_0, x \sim p(\cdot|x_0)}\left[\sum_{y \neq x} w_{xy}\left(s_\theta(x)_y - \frac{p(y|x_0)}{p(x|x_0)}\log s_\theta(x)_y\right) + K\left(\frac{p(y|x_0)}{p(x|x_0)}\right)\right] \quad (11)$$

$\mathcal{L}_{\text{DSE}}$ is scalable since it only requires the evaluation of one $s_\theta$, namely $s_\theta(x)$. It is also particularly suitable for discrete diffusion since the intermediate densities $p_t$ are all perturbation of the base $p_0$. In particular, for SEDD, we sample data points $x_0 \sim p_{\text{data}}$, perturb with the forward diffusion transition $p_{t|0}(\cdot|x_0)$ to sample $p_t$, and then train with $\mathcal{L}_{\text{DSE}}$ using the transition densities $p_{t|0}(\cdot|x_0)$.

## 3.2 LIKELIHOOD BOUND FOR SCORE ENTROPY DISCRETE DIFFUSION

**Fourth,** the score entropy can be used to define an ELBO for likelihood-based training and evaluation.

**Definition 3.5.** *For our time dependent score network $s_\theta(\cdot, t)$, the parameterized reverse matrix is*
$$\overline{Q}_t^\theta(y, x) = \begin{cases} s_\theta(x,t)_y Q_t(x,y) & x \neq y \\ -\sum_{z \neq x} \overline{Q}_t(z,x) & x = y \end{cases} \text{ found by replacing the ground truth scores in Equation 7.}$$
*Our parameterized densities $p_t^\theta$ thus satisfy the following differential equation:*

$$\frac{dp_{T-t}^\theta}{dt} = \overline{Q}_{T-t}^\theta p_{T-t}^\theta \quad p_T^\theta = \pi \approx p_T \tag{12}$$

The log likelihood of data points can be bounded with an ELBO that depends only on the rate matrices (Campbell et al., 2022). Interestingly, this becomes our score entropy loss:

**Theorem 3.6** (Likelihood Training and Evaluation). *For the diffusion and forward probabilities defined above, we can upper bound the log-likelihood of individual data points*

$$-\log p_0^\theta(x_0) \leq \mathcal{L}_{\text{DWDSE}}(x_0) + D_{KL}(p_{T|0}(\cdot|x_0) \parallel \pi) \tag{13}$$

*where $\mathcal{L}_{\text{DWDSE}}(x_0)$ is the **diffusion weighted denoising score entropy** for data point $x_0$*

$$\int_0^T \mathbb{E}_{x_t \sim p_{t|0}(\cdot|x_0)} \sum_{y \neq x_t} Q_t(x_t, y) \left( s_\theta(x_t, t)_y - \frac{p_{t|0}(y|x_0)}{p_{t|0}(x_t|x_0)} \log s_\theta(x_t, t)_y + K \left( \frac{p_{t|0}(y|x_0)}{p_{t|0}(x_t|x_0)} \right) \right) dt \tag{14}$$

This means that, with a particular diffusion-based weighting scheme for $\mathcal{L}_{\text{DSE}}$ in the form of $\mathcal{L}_{\text{DWDSE}}$, our original training setup becomes maximizes likelihood training. We an also report an upper bound on $-\log p_0^\theta(x_0)$ for evaluation purposes.

## 3.3 PRACTICAL IMPLEMENTATION FOR LANGUAGE MODELING

**Fifth,** the score entropy can be scaled to high dimensional tasks.

In practice, our set $\{1, \ldots, N\}$ factorizes into sequences $\{1, \ldots, n\}^d$ (e.g. sentences of tokens or image pixel values) $\mathbf{x} = x^1 \ldots x^d$. To work with this factorization, our transition matrix $Q_t^{\text{seq}}$ instead perturbs tokens independently with a matrix $Q_t^{\text{token}}$ acting on each component $x^i \in \{1, \ldots, n\}$. The transition densities directly factorizes

$$p_{t|0}(\mathbf{y}|\mathbf{x}) = \prod_{i=1}^d p_{t|0}^{\text{token}}(y^i|x^i) \quad \frac{dp_t^{\text{token}}}{dt} = Q_t^{\text{token}} p_t^{\text{token}} \tag{15}$$

The full sequence transition matrix $Q_t^{\text{seq}}$ is mostly $0$ except when the indexing sequences differ at one position (e.g. $x^1 \ldots x^i \ldots x^d$ and $x^1 \ldots \widehat{x}^i \ldots x^d$ ). By the loss weighting of $\mathcal{L}_{\text{DWDSE}}$, we only need to model and learn the ratios between two sequences that differ at one position. This can be modeled similar to non-autoregressive language modeling tasks with our score network $s_\theta(\cdot, t) : \{1, \ldots, n\}^d \to \mathbb{R}^{d \times n}$ where

$$(s_\theta(x^1 \ldots x^i \ldots x^d, t))_{i,y} \approx \frac{p_t(x^1 \ldots \widehat{x}^i \ldots x^d)}{p_t(x^1 \ldots x^i \ldots x^d)} \tag{16}$$

To efficiently compute the other parts of $\mathcal{L}_{\text{DWDSE}}$, we need to compute the (token) forward transitions $p_{t|0}(x_t^j|x_0^i)$ We follow previous convention and define $Q_t^{\text{token}} = \sigma(t)Q$ for a fixed graph Laplacian-based $Q$ and a noise level $\sigma$. If we define $\overline{\sigma}(t) = \int_0^t \sigma(s)ds$ as the total noise level, the forward densities thus satisfy

$$p_t^{\text{token}} = \exp\left(\overline{\sigma}(t)Q\right) p_0^{\text{token}} \quad p_{t|0}^{\text{token}}(\cdot|x) = x\text{-th column of } \exp\left(\overline{\sigma}(t) \cdot Q\right) \tag{17}$$

To scale to GPT-2 experiments (where $d = 1024$, $n = 50257$, and the batch size is $64$ per GPU), there are some practical consequences that render most $Q$ unusable. In particular, one is not able to store all edge weights $Q_t(i, j)$ (since this takes around 20 GB of GPU memory and is extremely slow to access) used to compute $\mathcal{L}_{\text{DWDSE}}$. Furthermore, one must be able to compute the columns $\exp(\overline{\sigma}(t) \cdot Q)$ to get the transition ratios, but again one can't directly store all of them in memory.

We use two standard matrices with special structures that sidestep the above issues (Austin et al., 2021; Campbell et al., 2022). They arise, respectively, from considering a simplicial graph structure and the MASK token used in models such as BERT (Devlin et al., 2019):

$$Q^{\text{uniform}} = \mathbb{1} - N\mathbb{I}_N \in \mathbb{R}^{N \times N} \quad Q^{\text{absorb}} = \begin{bmatrix} -1 & 0 & \cdots & 0 & 0 \\ 0 & -1 & \cdots & 0 & 0 \\ \vdots & \vdots & \ddots & \vdots & \vdots \\ 0 & 0 & \cdots & -1 & 0 \\ 1 & 1 & \cdots & 1 & 0 \end{bmatrix} \in \mathbb{R}^{(N+1) \times (N+1)} \quad (18)$$

Notably, with such a structured $Q$, one can compute all the values in $\mathcal{L}_{\text{DWDSE}}$ quickly without much memory overhead. As such, our score entropy training iterations are about as quick and use a similar amount of memory as a standard autoregressive model training iteration.

## 4 SIMULATING REVERSE DIFFUSION WITH CONCRETE SCORES

Given our scores $s_\theta$, we now derive various strategies for simulating the (factorized) reverse diffusion process $\mathbf{x}_t = x_t^1 x_t^2 \ldots x_t^d \sim p_t$. Notably, the additional information that we gain from $s_\theta$ being an approximate ratio of $p_t$ can be used to enhance the sampling process.

### 4.1 TIME-REVERSAL STRATEGIES

To simulate the diffusion in Definition 3.5, one may be tempted to use the Euler strategy from Equation 6. However, as noted in Campbell et al. (2022), this is inefficient because the structure of $Q_t^{\text{seq}}$ means we can only alter one token per step. Instead, a natural alternative has been to use $\tau$-leaping (Gillespie, 2001) to simultaneously step through all states at once $p_{t-\Delta t|t}(\mathbf{x}_{t-\Delta t}|\mathbf{x}_t) = \prod_{i=1}^n p^i(x_{t-\Delta t}^i|\mathbf{x}_t^i)$

$$p^i(x_{t-\Delta t}^i|\mathbf{x}_t) = \begin{cases} \Delta t \cdot Q_t^{\text{token}}(x_t^i, x_{t-\Delta t}^i)s_\theta(\mathbf{x}_t, t)_{i,x_{t-\Delta t}^i} & x_{t-\Delta t}^i \neq x_t^i \\ 1 - \Delta t \sum_{y \neq x_t^i} Q_t^{\text{token}}(x_t^i, y)s_\theta(\mathbf{x}_t, t)_{i,y} & x_{t-\Delta t}^i = x_t^i \end{cases} \quad (19)$$

where the probabilities are clipped and normalized to account for discretization error. However, this procedure is agnostic to the probabilistic information of $s_\theta$. As an alternative, we introduce a discrete analogue of the famous Tweedie's theorem (Efron, 2011):

**Theorem 4.1** (Discrete Tweedie's Theorem). *Suppose that a distribution $p_t$ is a perturbation of a base distribution $p_{t-\epsilon}$ with a diffusion matrix $\exp(\overline{\sigma}Q)$. Then the (exact) reverse transition is given by*

$$p_{t-\epsilon|t}^\theta(x_{t-\epsilon}|x_t) = \left( \exp(-\overline{\sigma}Q) \left[ \frac{p_t(y))}{p_t(x_t)} \right]_{i=1}^N \right)_{x_0} \exp(\overline{\sigma}Q)_{x_t,x_0} \quad (20)$$

Note that this denoising scheme can not be directly applied to our language modeling task. In particular, we are not modeling the ratios between any two sequences, as otherwise this would allow us to generate from $p_T$ to $p_0$ in only one step. However, we can use this intuition to build an Tweedie $\tau$-leaping update:

$$p(x_{t-\Delta t}^i|\mathbf{x}_t) = \left( \exp(\overline{\sigma}_{\Delta t}(t)Q)s_\theta(\mathbf{x}_t, t)_i \right)_{x_{t-\Delta t}^i} \exp(\overline{\sigma}_{\Delta t}(t)Q)_{x_t^i,x_{t-\Delta t}^i} \quad (21)$$

$$\overline{\sigma}_{\Delta t}(t) = \overline{\sigma}(t) - \overline{\sigma}(t - \Delta t) \quad (22)$$

Interestingly, this is optimal when one forces the $\tau$-leaping scheme. In particular the key feature is that the overall transition factorizes by each token and that each token transition probability assumes other dimensions do not change.

**Theorem 4.2** (Tweedie $\tau$-leaping). *Let $p_{t-\Delta t|t}^{\theta^*}$ be the update rule defined by Equation 21 when our score function $s_\theta^*$ is learned perfectly. Then, $p_{t-\Delta t|t}^{\theta^*}$ minimizes the KL divergence $D_{\text{KL}}\left( p_{t-\Delta t|t}(\mathbf{x}_{t-\Delta t}|\mathbf{x}_t) \parallel p_{t-\Delta t|t}^\theta(\mathbf{x}_{t-\Delta t}|\mathbf{x}_t) \right)$ when $p_{t-\Delta t|t}^\theta$ factorize by dimension and each token transition does not depend on whether other dimensions are altered.*

### 4.2 ARBITRARY PROMPTING AND INFILLING

Our concrete score can be used to enable greater control over the generative process. In particular, we consider the infilling problem defined by the conditional probabilities

$$p_t(\mathbf{x}^\Omega | \mathbf{x}^{\overline{\Omega}} = \mathbf{y}) \quad \Omega \text{ unfilled indices} \quad \overline{\Omega} \text{ already filled indices.} \tag{23}$$

for example, a standard autoregressive conditional generation would have $\overline{\Omega} = \{1, 2, \ldots, c\}$ and $\Omega = \{c + 1, c + 2, \ldots, d\}$. By Bayes' rule, the conditional scores can be recovered exactly from the unconditional score.

$$\frac{p_t(\mathbf{x}^\Omega = \mathbf{z}' | \mathbf{x}^{\overline{\Omega}} = \mathbf{y})}{p_t(\mathbf{x}^\Omega = \mathbf{z} | \mathbf{x}^{\overline{\Omega}} = \mathbf{y})} = \frac{p_t(\mathbf{x} = \mathbf{z}' \oplus \mathbf{y})}{p_t(\mathbf{x} = \mathbf{z} \oplus \mathbf{y})} \quad \oplus \text{ is concatenation} \tag{24}$$

We can therefore approximate the relevant ratios (namely those with with one changed index) with our vanilla score function $s_\theta$, which justifies the following sampling procedure

$$\mathbf{x}_{t-\Delta t} = \text{proj}_{\overline{\Omega} \to \mathbf{y}}(\text{sample}_{t-\Delta t}(\mathbf{x}_t, s_\theta)) \tag{25}$$

(i.e. projecting the known indices after each score function-based step/only changing the unknown indices). In principle, we can also bound the likelihoods using Theorem 3.6. However, one major problem is that the probabilities $p_t(\mathbf{z} \oplus \mathbf{y})$ may be low, making it hard to learn $s_\theta(\mathbf{z} \oplus \mathbf{y}, t)$. If this is the case, we can follow previous work and approximate this with $s_\theta(\mathbf{z} \oplus \mathbf{y}(t), t)$ (Song et al., 2021b).

## 5 EXPERIMENTS

We now empirically validate that our score entropy discrete diffusion (SEDD) model can compete with existing large-scale autoregressive models, namely GPT-2 (Radford et al., 2019).

### 5.1 MODEL AND TRAINING SETUP

Our model is a standard encoder-only transformer architecture (Vaswani et al., 2017) similar to standard masked language models (Devlin et al., 2019). However, our model incorporates the time conditioning method from Peebles & Xie (2023) and uses rotary instead of positional encodings (Su et al., 2021). Instead of outputting $s_\theta$ directly, we instead output $\log s_\theta$ to maintain positivity without clipping the output or gradients.

We report results for both the uniform and absorbing token matrices $Q^{\text{uniform}}$ and $Q^{\text{absorb}}$. For the absorbing transition for reported perplexities, we use a log-linear noise schedule that will mask out $\tau \sim U([0, d])$ tokens. For all other experiments/generations, we used a geometric noise schedule that interpolates between $10^{-5}$ and 20. Outside of this, we did not systemically explore noise schedules or alternative loss weighting, although these will most likely improve sample perplexity and generation (as is commonly seen for continuous diffusion).

We train on OpenWebText (Gokaslan & Cohen, 2019), an open source recreation of the WebText dataset used for training GPT-2 (in practice, this results in almost no difference for final models). We matched the architecture sizes of GPT-2, although our models have slightly more non parameters ($\approx 5 - 10\%$) due to time conditioning. We train for 400k iterations of batch size 512, typical for open source GPT-2 scripts (Dao et al., 2022). Further details are given in Appendix B.

### 5.2 PERPLEXITY SCORE COMPARISON

We follow GPT-2 and report zero-shot perplexities on the LAMBADA, WikiText2, PTB, WikiText103, and 1 Billion Words datasets. We recompute baseline likelihoods for all datasets except 1BW, where we encountered unexpected behavior with the public implementations. Our likelihood computation changes from the original setting since use different splits and evaluate unconditional likelihoods (ie without a sliding window). This results in slightly higher perplexities for GPT-2 than originally reported, although this is minor on most datasets.

Our results are reported in Table 1. Our absorbing transition models effectively match the performance of GPT-2, as their perplexities are commonly within a $+10\%$. Prior work (Song et al., 2021a) has shown that this is around the gap between exact likelihoods and the variational bounded for continuous space diffusion, although it is unknown if this holds true for discrete spaces. Furhtermore, the uniform transition models consistently underperforms.

|  | LAMBADA | WikiText2 | PTB | WikiText103 | 1BW |
|---|---|---|---|---|---|
| GPT-2-small | **45.04** | **42.43** | 138.43 | **41.60** | **75.20**\* |
| SEDD-small Absorb | ≤52.21 | ≤44.75 | ≤**130.49** | ≤43.14 | ≤80.70 |
| SEDD-small Uniform | ≤66.94 | ≤55.88 | ≤144.88 | ≤53.90 | ≤100.86 |
| GPT-2-medium | **35.66** | **31.80** | 123.14 | **31.39** | **55.72**\* |
| SEDD-medium Absorb | ≤44.60 | ≤34.85 | ≤**93.26** | ≤32.97 | ≤67.91 |
| SEDD-medium Uniform | ≤51.14 | ≤39.79 | ≤100.58 | ≤37.69 | ≤79.26 |

Table 1: **Zero-shot unconditional perplexity (lower is better) on a variety of datasets.** For a fixed model size, the best perplexity is **bolded** and ELBO bounds that fall within the variational error of $+10\%$ are underlined. Our score entropy discrete diffusion (SEDD) model with absorbing transition almost matches the performance of GPT-2, although the uniform transition lags behind.

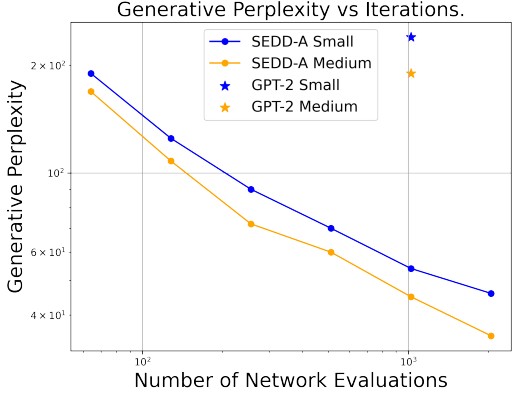

| | |
|---|---|
| **GPT-2** | Members of the prefabricated surplus yard placemat board of Metrolinx designated reserved land located next to Vectverified... |
| **SEDD-A** | As Jeff Romer recently wrote, "The economy has now reached a corner - 64% of household wealth and 80% of wealth goes to credit cards because of government austerity ... |
| **SEDD-U** | The pledge itself is an offer from the government, but the oil panhandlers is taking some of the proposed cost to the system of utilities in place ... |

(a) Generative Perplexity vs. Sampling Iterations.   (b) Generated Text (small models, non-picked)

Figure 2: **Evaluation of generated text.** We compare the ancestral sampling techniques of autoregressive and diffusion models (GPT2 vs SEDD). Our SEDD Absorb models consistently outperform GPT2, interpolating between a $16\times$ speedup and a $5\times$ improvement based on the chosen step size. The generated text reflects this improved generation capability. Additional samples in Appendix C.4

## 5.3 SAMPLE QUALITY COMPARISON

We generate unconditional samples of length $1024$ and report the generative perplexity (as measured by GPT-2 Large). For all methods, we use the analytic sampling method for GPT-2 and our reverse diffusion sampler for SEDD to fairly compare the modeled probability distributions. Note that other commonly used autoregressive sampling methods (ie beam search or nucleus sampling) don't sample from the true distribution and are adversarial against our evaluation objective since they are made to explicitly decrease the perplexity of the generated sequences (Freitag & Al-Onaizan, 2017; Holtzman et al., 2020). Our results are shown in Figure 2: SEDD with uniform transition consistently outperforms GPT-2 while creating a log-log linear pareto frontier between sampling steps and generation quality. Additional results and samples are given in Appendix C.

## 5.4 ARBITRARY INFILLING

We showcase our ability to condition our generation with inputs at arbitrary location. Our results are shown in Table 2. Additional samples are given in Appendix C.4.

## 6 RELATED WORK

We briefly highlight other diffusion language methods and compare against them in Figure 3. SEDD produces significantly better perplexities and generates better samples than existing diffusion methods. Note that these models are only trained for 60k iterations, which is why the SEDD and GPT-2 results are higher than in our Figure 2. Full details are given in Appendix C.2.

| |
|---|
| A bow and arrow is a traditional weapon used by penury Englishmen. The gun shoots into water, starvation and thunder centuries after short-range weapons were built. The weapon is the focus of a new exhibition Dr Tom Fellow, from Pcock, is curator of objects at the History Museum in Oxford. ... |
| ... seems to have known skydiving is a fun sport that exists, in other words, subliminally like climbing the feeling is exhilarating. Watson is beginning to wonder, as their conversation on it continues, why not. "One thing springs to mind," she says. ... |
| ... with significantly lower skin infections. Also this year a Franklin study published a report that found that with more use of reliable medical data, monthly changes following a nutritional boost could have a devastating stay in school kids. |
| ... as if he could have been erred, (Donald Trump and Hillary Clinton started to change their position. Some, as Tom and Perez mentioned, were good specifics, such as where they have a letter the FFP agents give their way to pass to offsetting ... |

Table 2: **Conditionally Generated Text.** Prompt tokens are given in blue. Our model is able to generate meaningful text with prompt tokens in the front, the end, the middle, or even split up. First two samples are generated by SEDD medium, while the last two are from SEDD small. Additional samples are given in Appendix C.4.

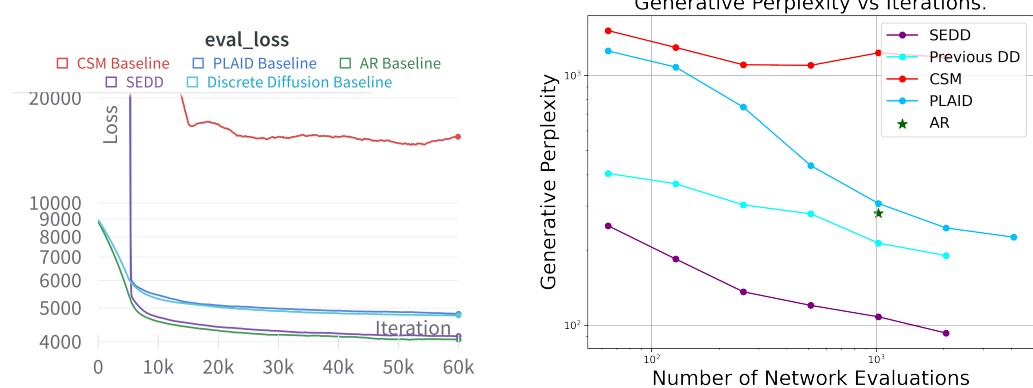

Figure 3: **Ablation of Prior Language Diffusion Models (at 60k training iterations).** Left: validation ELBO curves for our method, an autoregressive model, and prior diffusion models. Prior diffusions have $> 2\times$ worse perplexities, and CSM is significantly worse. Right: generative perplexities of the models. Our method produces noticeably better samples than baselines methods.

**Previous Discrete Diffusion.** Previous discrete diffusion works draw inspiration from Ho et al. (2020) and parameterize $\overline{Q}$ with the "mean" probability distribution $p_{0|t}(\mathbf{x}_0|\mathbf{x}_t)$ (Austin et al., 2021; Hoogeboom et al., 2022; Campbell et al., 2022; He et al., 2022). However, these models typically perform anywhere from 2 to 4 times worse than autoregressive models on non-toy language modeling datasets like 1BW (Austin et al., 2021; He et al., 2022). This is corroborated in Figure 3.

**Continuous Diffusion.** Several works embed the tokens into Euclidean space and learn a continuous diffusion model (Li et al., 2022; Dieleman et al., 2022; Gulrajani & Hashimoto, 2023). When compared with autoregressive models, these methods require $10\times$ the parameters for similar perplexities (Gulrajani & Hashimoto, 2023) and a comparable (if not more) amount of sampling steps to generate high quality samples. This is again shown in Figure 3.

## 7 CONCLUSION

We have introduced score entropy discrete diffusion (SEDD) models, a new class of discrete diffusion model that is parameterized by the concrete score and can be learned efficiently with our novel score entropy loss. SEDD achieves competitive performance in a direct head-to-head with GPT-2, almost matching likelihoods while generating higher quality samples with more control options. We hope that future work can build off of our framework to define diffusion model alternatives to the modern autoregressive language modeling paradigm.

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

## 8 OTHER RELATED WORKS

**Ratio Matching Conditionals.** Sun et al. (2023) notes that the ratios parameterize the reverse distribution but instead optimizes with the ratio matching loss (Hyvärinen, 2007), which parameterizes the conditional distributions $p_t(x^i|(x^j)_{j\neq i})$. This setup differs greatly because one must construct specialized architectures to make sure $x^i$ does not affect $p(x^i)$ (Chen & Duvenaud, 2019), which causes this method to typically perform worse than mean prediction. Because of these reasons (specialized architecture, worse results than mean prediction, method did not evaluate for language modeling) we do not include this in our ablation study.

## 9 FUTURE WORK

Despite our contribution, much work remains before discrete diffusion models can truly rival modern autoregressive models. For example, the effects of scaling on performance (Hoffmann et al., 2022) are unexplored, and diffusion models are limited since they generate full length outputs. Furthermore, existing systems level improvements such as the KV cache can cut into our algorithmic speedup (KV-cache autoregressive modeling has a similar sampling runtime to our sampling method using $64$ iterations).

However, our work takes the crucial first step in showcasing competitive viability and demonstrating tangible benefits for language diffusion modeling. Therefore, we remain optimistic that future work in this direction can rival the domination of autoregressive models. By generalizing improvements in the continuous diffusion model framework, future work could tune the noise schedule to further improve generation quality (Dhariwal & Nichol, 2021), leverage score-based controllability methods Ho (2022), or further reduce the number of sampling steps (Song et al., 2023).

## A PROOF OF MAIN RESULTS

*Proof of Prop 3.2.* Given infinite samples, the loss becomes equivalent to minimizing

$$\min_\theta \sum_{x,y\neq x} p(x) w_{xy} \left( s_\theta(x)_y - \frac{p(y)}{p(x)} \log s_\theta(x)_y \right) \qquad (26)$$

where we have removed constants not depending on $\theta$. This is minimized when

$$s_\theta(x)_y - \frac{p(y)}{p(x)} \log s_\theta(x)_y \qquad (27)$$

is minimized for all $x, y$. Taking a derivative with respect to $s$ and setting to $0$, we see that this occurs when $s_\theta(x)_y = \frac{p(y)}{p(x)}$, which can be easily checked to be optimal as the function is convex as a function of $s$. One can check that the loss is $0$ at the minimum. $\qquad \square$

*Proof of Prop 3.3.* The trick is the categorical equivalent of the divergence theorem. In particular, we have

$$\mathbb{E}_{x\sim p} \sum_{y\neq x} \frac{p(y)}{p(x)} f(x,y) = \sum_{x,y:x\neq y} \frac{p(y)}{p(x)} p(x) f(x,y)$$

$$= \sum_{x,y:x\neq y} p(y) f(x,y)$$

$$= \mathbb{E}_{y\sim p} \sum_{x\neq y} f(x,y)$$

$$= \mathbb{E}_{x\sim p} \sum_{y\neq x} f(y,x)$$

for abitrary $f$. By setting $f(x, y) = w_{xy} \log s_\theta(x)_y$, we get that

$$\mathbb{E}_{x \sim p} \left[ \sum_{y \neq x} w_{xy} \left( s_\theta(x)_y - \frac{p(y)}{p(x)} \log s_\theta(x)_y + K \left( \frac{p(y)}{p(x)} \right) \right) \right]$$

$$= \mathbb{E}_{x \sim p} \left[ \sum_{y \neq x} w_{xy} s_\theta(x)_y - w_{yx} \log s_\theta(y)_x + w_{xy} K \left( \frac{p(y)}{p(x)} \right) \right]$$

which is the desired equivalent (as the last term does not depend on $\theta$). $\qquad\square$

*Proof of Thm 3.4.* This is similar to the same denoising variant for concrete score matching. We just need to show that the $\log s_\theta(x_t)_y \frac{p_t(y)}{p_t(x)}$ marginalizes out, since everything else does not change or is a constant.

$$\mathbb{E}_{x \sim p} \sum_{y \neq x} f(x, y) \frac{p(y)}{p(x)} = \sum_{y \neq x} f(x, y) p_t(y)$$

$$= \sum_{y \neq x} \sum_{x_0} f(x_t, y) p(y|x_0) p_0(x_0)$$

$$= \mathbb{E}_{x_0 \sim p_0} \sum_{y \neq x} f(x, y) \frac{p(y|x_0)}{p(x|x_0)} p(x|x_0)$$

$$= \mathbb{E}_{x_0 \sim p_0, x \sim p(\cdot|x_0)} \sum_{y \neq x} f(x, y) \frac{p(y|x_0)}{p(x|x_0)}$$

Applying this to our loss when $f(x, y) = w_{xy} \log s_\theta(x)_y$ gives us

$$\mathbb{E}_{x \sim p} \left[ \sum_{y \neq x} w_{xy} \left( s_\theta(x)_y - \frac{p(y)}{p(x)} \log s_\theta(x)_y + K \left( \frac{p(y)}{p(x)} \right) \right) \right]$$

$$= \mathbb{E}_{x \sim p} \left[ \sum_{y \neq x} w_{xy} \left( s_\theta(x)_y + K \left( \frac{p(y)}{p(x)} \right) \right) \right] - \mathbb{E}_{x_0 \sim p_0, x \sim p(\cdot|x_0)} \left[ \sum_{y \neq x} \frac{p(y|x_0)}{p(x|x_0)} w_{xy} \log s_\theta(x)_y \right]$$

$$= \mathbb{E}_{x_0 \sim p_0, x \sim p(\cdot|x_0)} \left[ w_{xy} \left( s_\theta(x)_y \frac{p(y|x_0)}{p(x|x_0)} \log s_\theta(x)_y + K \left( \frac{p(y)}{p(x)} \right) \right) \right]$$

$\qquad\square$

*Proof of Thm 3.6.* The full bound is given by

$$- \log p_0^\theta(x_0) \leq \mathcal{L}_{\text{DWDSE}}(x_0) + D_{\text{KL}}(p_{T|0}(\cdot|x_0) \| \pi) \tag{28}$$

where $\mathcal{L}_{\text{DWDSE}}$ is given by

$$\int_0^T \mathbb{E}_{x_t \sim p_{t|0}(\cdot|x_0)} \sum_{y \neq x_t} Q_t(x_t, y) \left( s_\theta(x_t, t)_y - \frac{p_{t|0}(y|x_0)}{p_{t|0}(x_t|x_0)} \log s_\theta(x, t)_y + K \left( \frac{p_{t|0}(y|x_0)}{p_{t|0}(x_t|x_0)} \right) \right) dt$$

Effectively, $\mathcal{L}_{\text{DWSDE}}$ is the path measure KL divergence (Campbell et al., 2022; Song et al., 2021a), and the proof follows similarly. In particular, we have that, by the data processing inequality

$$- \log p_0^\theta(x_0) = D_{\text{KL}}(\delta_{x_0} \| p_0^\theta) \leq D_{\text{KL}}(\mathbb{P}_{x_0} \| \mathbb{P}^\theta) \tag{29}$$

where $\mathbb{P}_{x_0}$ is the path measure for the reverse of the noising process applied to $\delta_{x_0}$ and $\mathbb{P}^\theta$ is the learned reverse process. Generally, we can replace $\delta_{x_0}$ with a more general data distribution $p_{\text{data}}$, with the computation remaining the same. We have,

$$D_{\text{KL}}(\mathbb{P}_{x_0} \| \mathbb{P}^\theta) \leq \mathbb{E}_{x_T \sim p_{T|0}(\cdot|x_0)} \left[ D_{\text{KL}}(\mathbb{P}_{x_0}(\cdot|x_T) \| \mathbb{P}^\theta(\cdot|x_T)) \right] + D_{\text{KL}}(p_{T|0}(\cdot|x_0) \| \pi) \tag{30}$$

We analyze the term $\mathbb{E}_{x_T} D_{\mathrm{KL}}(\mathbb{P}_{x_0}(\cdot|x_T) \parallel \mathbb{P}^\theta(\cdot|x_T))$, which we can compute by Dynkin's formula (Hanson, 2007; Campbell et al., 2022), which, similar to Girsanov's Theorem for standard SDEs (Øksendal, 1987), allows one to compute the change in measure. In particular, by applying Theorem 7.1 of Hanson (2007) with degenerate SDE coefficients, we find the expectation to be given explicitly by

$$\int_0^T \mathbb{E}_{x_t \sim p_{t|0}(\cdot|x_0)} \sum_{y \neq x_t} \overline{Q}_t^\theta(y, x_t) - Q_t(y, x_t) \log(\overline{Q}_t^\theta(x_t, y)) \tag{31}$$

$$+ Q_t(y, x_t) \log Q_t(y, x_t) + Q_t(x_t, y) K\left(\frac{p_{t|0}(y|x_0)}{p_{t|0}(x_t|x_0)}\right) dt \tag{32}$$

Since our reverse rate matrices $\overline{Q}_t^\theta$ are parameterized with $s_\theta$, we can simplify the above to

$$\int_0^T \mathbb{E}_{x_t \sim p_{t|0}(\cdot|x_0)} \sum_{y \neq x_t} Q_t(x_t, y) \left( s_\theta(x_t, t)_y + K\left(\frac{p_{t|0}(y|x_0)}{p_{t|0}(x_t|x_0)}\right) \right) - Q_t(y, x_t) \log s_\theta(y, t)_{x_t} dt \tag{33}$$

To finalize, we simply note that the summation over $Q(y, x_t) \log(s_\theta(y, t)_{x_t})$ can be simplified with the (reverse of) the trick used for proving 3.3.

$$\mathbb{E}_{x_t \sim p_{t|0}(\cdot|x_0)} \sum_{y \neq x_t} Q(y, x_t) \log s_\theta(y)_{x_t} = \sum_{x_t, y \neq x_t} p_{t|0}(x_t|x_0) Q(y, x_t) \log s_\theta(y)_{x_t} \tag{34}$$

$$= \mathbb{E}_{y \sim p_{t|0}(\cdot|x_0)} \frac{p_{t|0}(x_t|x_0)}{p_{t|0}(y|x_0)} Q(y, x_t) \log s_\theta(y)_{x_t} \tag{35}$$

$$= \mathbb{E}_{x_t \sim p_{t|0}(\cdot|x_0)} \frac{p_{t|0}(y|x_0)}{p_{t|0}(x_t|x_0)} Q(x_t, y) \log s_\theta(x_t)_y \tag{36}$$

where the last line is just a permutation of the notation of $x_t$ and $y$. As such, we get the desired loss

$$\int_0^T \mathbb{E}_{x_t \sim p_{t|0}(\cdot|x_0)} \sum_{y \neq x_t} Q_t(x_t, y) \left( s_\theta(x_t, t)_y - \frac{p_{t|0}(y|x_0)}{p_{t|0}(x_t|x_0)} \log s_\theta(x, t)_y + K\left(\frac{p_{t|0}(y|x_0)}{p_{t|0}(x_t|x_0)}\right) \right) dt$$

$$\square$$

*Proof of Thm 4.1.* This can be shown by Bayes' rule:

$$p_{0|t}(x_0|x_t) = \frac{p_{t|0}(x_t|x_0) p_0(x_0)}{p_t(x_t)} = p_{t|0}(x_t|x_0) \frac{p_0(x_0)}{p_t(x_t)} \tag{37}$$

We have $p_0 = \exp(-\sigma Q) p_t$ and $p_{t|0}(x_t|x_0) = \exp(\sigma Q)_{x_t, x_0}$, so the theorem follows. $\square$

*Proof of Thm 4.2.* Using our factorization assumption we get that

$$D_{\mathrm{KL}}\left(p_{t-\Delta t|t}(\mathbf{x}_{t-\Delta t}|\mathbf{x}_t) \parallel p_{t-\Delta t|t}^\theta(\mathbf{x}_{t-\Delta t}|\mathbf{x}_t)\right) \tag{38}$$

$$= -\sum_{i=1}^d \mathbb{E}_{\mathbf{x}_{t-\Delta t} \sim p_{t-\Delta t|t}(\mathbf{x}_{t-\Delta t}|\mathbf{x}_t)} \left[\log p_{t-\Delta t|t}^\theta(x_{t-\Delta t}^i|\mathbf{x}_t)\right] + C \tag{39}$$

where $C$ is a constant independent of $\theta$. We simply need to minimize the following cross entropy loss for each $i$

$$-\mathbb{E}_{\mathbf{x}_{t-\Delta t} \sim p_{t-\Delta t|t}(\mathbf{x}_{t-\Delta t}|\mathbf{x}_t)} \left[\log p_{t-\Delta t|t}^\theta(x_{t-\Delta t}^i|\mathbf{x}_t)\right] \tag{40}$$

Our $\tau$-leaping condition implies that our transition assumes no change in other dimensions, so in particular $p_{t-\Delta t}^i(x_{t-\Delta t}^i|\mathbf{x}_t) = p_{t-\Delta t|t}^\theta(x_t^1 \ldots x_{t-\Delta t}^i \ldots x_t^d|\mathbf{x}_t)$. By the standard properties of cross entropy, this is minimized when $p_{t-\Delta t|t}^\theta(x_t^1 \ldots x_{t-\Delta t}^i \ldots x_t^d|\mathbf{x}_t) = p_{t-\Delta t|t}(\mathbf{x}_{t-\Delta t}|\mathbf{x}_t)$. This equality follows directly from Thm 4.1. $\square$

## B  ADDITIONAL EXPERIMENTAL DETAILS

### B.1  DIFFUSION DETAILS

The geometric noise distribution is $\overline{\sigma}(t) = \sigma_{\min}^{1-t}\sigma_{\max}^t$. The log linear noise schedule is $\overline{\sigma}(t) = -\log(1 - (1 - \epsilon t))$ for some small epsilon for numerical stability as $t \to 1$, commonly $10^{-3}$ or $10^{-4}$. These noise schedules were chosen such that the prior loss $D_{\mathrm{KL}}(p_{T|0}(\cdot x_0) \parallel \pi)$ and the approximation of $p_{\mathrm{data}}$ with $p_{\overline{\sigma}(0)}$ are negligible. We typically scale the uniform transition matrix down by $\frac{1}{N}$ and take $\pi$ to be uniform. For the absorbing state, we take $\pi$ to be the MASK state with some leakage of probability to the non-MASK state (to avoid $\inf$ KL divergence, although this is negligible and is not used for generation in practice).

### B.2  MODEL DETAILS

Our model train with flash attention (Dao et al., 2022) with fused kernels wherever applicable. We also use the adaLN-zero time information network of (Peebles & Xie, 2023) with 128 hidden dimension. Following previous work, we parameterize the network with the total noise level instead of the time $t$.

SEDD models have the same hidden dimensions, number of blocks, and number of heads as their corresponding GPT-2 models. However, SEDD models also use a separate word embedding matrix and output matrix. In total, SEDD small and SEDD medium have around 90M parameters and 320M non embedding parameters respectively (compared to GPT-2 small 86M and GPT-2 medium 304M non-embedding parameters respectively).

### B.3  TRAINING DETAILS

All models were trained with a batch size of 512 and trained with a learning rate of $3 \times 10^{-4}$. We clip our gradient norm to 1 and have a linear warmup schedule for the first 2000 iterations. In accordance with previous work, we report our results using a 0.9999 EMA.

We trained on nodes of 8 A100 80GB or 16 A100 40GB GPUs, using gradient accumulation when our batch size did not fit into memory (as is the case for SEDD medium).

### B.4  HYPERPARAMETER SEARCH

We did not do a hyperparameter or achitecture search. Our hyperparameters were chosen for convenience purposes (e.g. the architecture was taken from DDiT (Peebles & Xie, 2023), but we use rotary embeddings since they come included in flash attention (Dao et al., 2022)) or were naturally lifted from previous training recipes (e.g. the $3 \times 10^{-4}$ learning rate, 0.9999 EMA).

### B.5  EVALUATION DETAILS

We randomly sample with 1000 timesteps to Monte Carlo estimate our likelihoods. We use invertible tokenizers, as is customary for GPT-2 experiments. We report results on the test set for all datasets besides WikiText02, where we report on the train set since WikiText02 and WikiText103 share the same test set.

For the generative perplexity results, we generate 1000 samples. For the GPT-2 model, since there must be some conditional generation, we simply fed the model a random initial token.

For conditional generation, we used the SEDD with absorbing transition matrix.

## C  ADDITIONAL EXPERIMENTAL RESULTS

### C.1  TOY EXPERIMENT ON TEXT8

We validate that our method works better on simple datasets, in particular the text8 dataset. We train our SEDD model according to the training scheme (architecture size and tokenization) of Austin et al. (2021). We achieve a $1.41$ bpc (bits per character) for absorbing diffusion (improving upon their

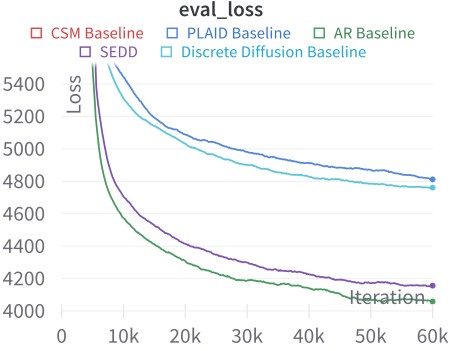 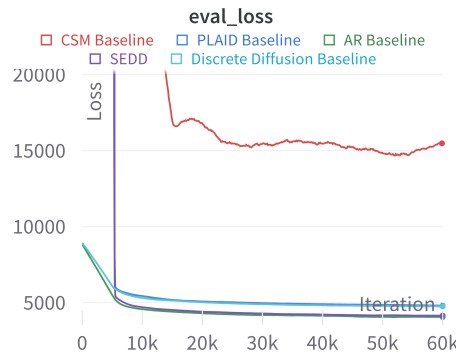

Figure 4: **Loss Curves of Various Language Diffusion Models. Left: zoomed in. Right: zoomed out to include Concrete Score Matching.** We test our score entropy (SEDD) against previous discrete diffusion methods (Discrete Diffusion Baseline), continuous diffusion (PLAID), and concrete score matching (CSM). An autoregressive GPT-2 baseline is included for comparison (AR). We see that our method vastly outperforms previous language diffusion methods/losses.

result of $1.47$ when trained with the likelihoods and $1.44$ with an augmented loss). We also achieve a $1.44$ bpc for the uniform noise, significantly improving upon their reported $1.61$. We believe this gap in the uniform noise case is caused by our parameterization of the concrete score. While the absorbing state result is not a big gap over previous methods, we will now see that it makes a big difference for larger scale experiments, possibly because of the underlying data complexity or the large vocabulary size.

## C.2 ABLATION OF PREVIOUS METHODS

In this section, we ablate previous methods, showing that our score parameterization and training method produce a substantial improvement even when controlling for architecture.

We consider the mean parameterized discrete diffusion method of Austin et al. (2021); Campbell et al. (2022), the Concrete Score Matching objective from Meng et al. (2022), and the continuous space diffusion method given by PLAID (Gulrajani & Hashimoto, 2023). We adapt these methods to our test task (training on OpenWebText with the GPT-2 tokenizer) and train using our small-DDiT backbone (Peebles & Xie, 2023) for all models and use the same hyperparameters (except for PLAID, where we lift the hyperparameters from the original code). For the discrete diffusion methods, we train with the absorbing noise schedule. We also include results for an autoregressive GPT-2 as a baseline comparison. We train and evaluate using the ELBO for all likelihood-based methods. Since CSM can't be trained with an ELBO, we simply replaced our score entropy term with the corresponding concrete score matching term during training (although we still report the ELBO for consistency). This matches the proposed "graph-weighting" proposed by Meng et al. (2022) and is the most natural choice.

We report the validation log-likelihood (roughly equivalent to $1024\times$ log perplexity) on the Wiki-Text103 validation test set during training (running average of 100 steps). Our results are shown in Figure 4. A summary of our results:

1. Our score entropy is slightly worse than the autoregressive baseline, as expected from our final results.

2. Previous discrete diffusion methods is $\approx 0.6$ natural units worse than our score entropy training, which is roughly equivalent to $1.8\times$ higher perplexity.

3. Continuous diffusion is $\approx 0.7$ natural units worse than our score entropy training, roughly equivalent to $2\times$ higher perplexity.

4. Concrete score matching does not optimize well. The likelihood improves up to a certain point (which is $\approx 4\times$ our ELBO).

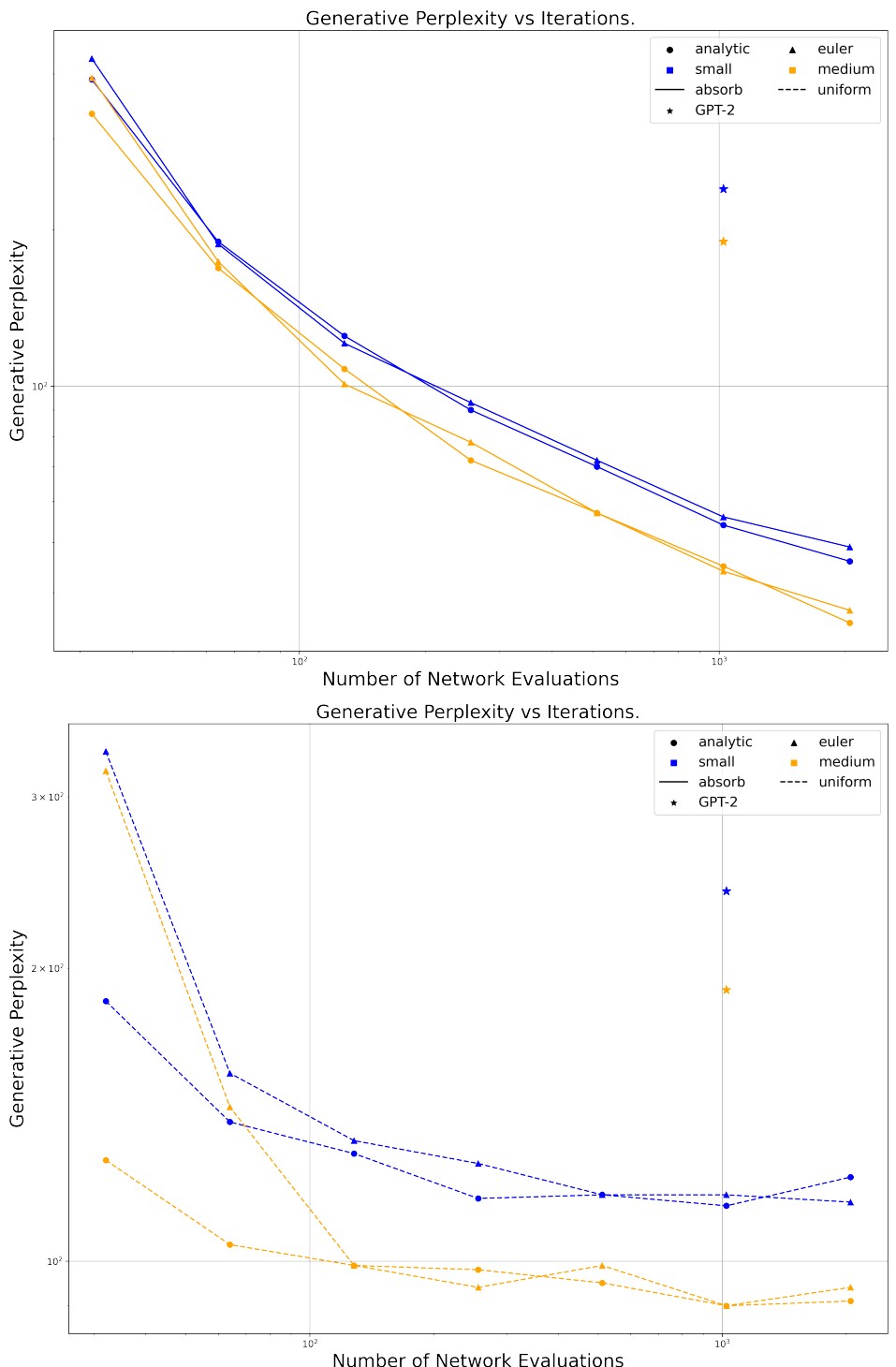

Figure 5: Generative Perplexity for SEDD absorbing (top) and uniform (bottom).

## C.3 FURTHER EVALUATION OF GENERATIVE PERPLEXITY

We further evaluate our generative perplexity for uniform models as well as different sampling schemes (analytic sampling based on Tweedie's vs Euler sampling based off of reverse diffusion). Results are shown in Figure 5. Generally, we find that uniform does not produce the same linear

tradeoff curve as absorbing (most likely due to a bottleneck in generation quality). Futhermore, analytic generally outperforms Euler sampling, and this is a major factor for the uniform model.

## C.4 ADDITIONAL SAMPLES

; Koopong and Kozullo each received annual stipends of $500 for regular parking. Personnel and administration described how common illegal activities with their lawmakers were. Koopong had our neighbors respond as politically incorrect. Koltak adds, "People said their taxes were too high."

Other sidewalks that are not clean are clustered around stadiums and other venues that will (incidentally) become part of BB&T, they expressed joy. Bearing stones and flag-sporting players cheered following the signing. Players hit with the "Bill of Rights" signed by kits may claim PG&E shares in analysis fee like SBHR11 / glasses, lifestyle ebook for tattoo/sculpture projects and pirate rewards cards (12/25/00 for Subscription). Keiley, BA said there are six sitting Summons Vendors. Most of the other storefronts funnel $10,000 into real estate and work

work-times. The nature of Bose also inspired and painted a composite image aimed at encouraging the purchase of sitebursts. The Studio 15 tenant cried out as more business from Pulaski Grill, one of the city's premier club clubs, popped-up. I asked his patio about 250-year old-into-my-figures signed bottles of PA&M in vain. Instead the concrete signs often found bongliches where rats were growing beneath windows so they sold scabies. Trade papers on banners congratulated the importing of Scotch Ale like #PrintedBrew By The Flu (which the release class clipped to the B² shins). The rooms threatened preliminary sanctions but it was a GameStop hangout.

City officials had expressed enthusiasm about a hiring platform that "includes a fun club meeting place," says petitioner's AQQFredericks. They's the adjacent marijuana-hop. Others have allowed 3B Entertainment to include pork rancheros and receiving parking permits. Possibly AB 302 is coming. State Department of Licenses has ordered Pfizer to pay $67,000 tax exemption under the 1951 Marijuana Tax Act, he adds. Ajax responded with the same public-context query. Sierra Vista was secured to bear "branded" items of beer and asked to spend $200,000 to break it down to $10,000.

Brand Me Remembering Mac to not be Saul Bowmare I give you this. We'll see if she responds. "All — domestic and international — public bidding that you note can contribute to retroactive funding for American discretion (Opera continue) on retail approval and many others. Many doesn't post in the public grid." Begin parking off E. 93rd St., from woods behind Merush Correctional Facility onto E. 93rd St.*: "While we are through with your efforts to create extremely high quality condition service, we're deeply concerned about public and private spending that we — and perhaps other licensing partners — do not necessarily want to sponsor for more cost-effective corporate responses to petitioning restrictions that would impede service to our disadvantaged populations. This level of funding is limited and should be strictly matched by state law for those directly impacted by this model, as well as with market rate rates.

"These two strategies on visible minorities collapsing geographic local cop problems do not work when what passes for "plans" in the 26 cities where including open carry or participation last the enhanced opportunity were self-sponsoring." Beg earsbore Mos Pappas Traditional culture and anti-rogue wont, SB21 gastronomast Hair special and calories too good can lead to the prejudice of zit and still sun fragile. Anchored building are theorems for Jen Boulmerlin's ATVE

trobunal sponsorships where squeezed-out citizens would end up owing significant or all of their income in taxes. Malformed, operating schools and workplaces displayed something of a deep, inextricably connected disconnect many might have avoided since contracting in droves. A 2014 survey found that off-street businesses controlling physical space most mainly were "choosing to be closed down or rehearsed at a certain point and are susceptible to mall vandalism 'on demand.' Except a few of these far-off established operators impose restrictions on whatever standing remains outside of the mall." Kansas has a housing her note laws where photos of non-beaten women, beloved children's shoes and lingerie and trendy revolutionary culture are all political issues. Think Drive leaning with outside bounty on your heart. Tenants spent $30K on occupation benefits that failed to curb spine tics AND most eviction rules used Lucas Venturi docu schedules at his PlayPoint inner-site membership #280

Figure 6: GPT-2 Small Analytic Sampling. Unconditional

tired and half-mad about her eldest corner of life on her porch at 12. "My mother never lay outside her home," says Lamb's bihelson.

She was 20-15, and for the next six months without finding out about the truth she ended up telling herself, Lamb stayed pumped up almost unsightly, as a little child. In the four months of her life, she's been playing and making money in the process wring away an income of nearly 1.7 billion dollars.

It's not that long. Lamb stares despairingly at at least two people, amid pale-aged woodland and piles of campsites he now uses as an Atlanta Herald-Western reporter punching out his weary eyes.

"When many of these days went dark none of these forms could go forward without the offenders being high."

"At a few weeks my doctor came at home and had a camera and a book on the marijuana I was taking a young child," says Lamb, now named Sharon Schlessy. She believed that her mother's shaky health had gone on for a moment, but she couldn't do anything about it, but her stepmother lay dead in front of her. But nothing settled with some of her victims. Her mother shot her "every day with a bullet." Three weeks later, Lamb's came back again. "You cut down folks on trees," says the woman with her hair. "Every gun to drow on fruit trees right there was cheap, illegal, and on your own."

"While I was nine-year old Angela, there were 15 of my who came back in on-the-job and my best shot at life," she says. "Tiny took away substances in life, and your mother's life was financed by a small little gun we just bashed, and sometimes, I'd end up arguing in the closet with my mother where she killed her little crow [the tiny squirrel-nay] but couldn't catch anything." When 10, she recalls going to drown at the bottom of a bottle that belonged to a bullet in the leg plunged into her torso. "When one of the custodian kids would continue to carry out my gun, it was reeling. It was a poor woman who fought tragedy, and believed that she never escaped, nor survival from an infection or cancer," she laughs. Moreover, was the man Lamb and her friends worried about going wrong? Yes, and without. "Right now, I get passed and talked to back home," says 50-year-old.

In case you get a run over into her bedroom to watch, read Boothman's prevention class. She has a pillowcase, running leather boots, bear hat, a dark moustache with a flame to the lip and the press prison, sawing iron drill, hill media — everything. Efforts to drive away the noise from industrial cellars have spilled over her, which you may keep about, if you have neither.

The online processor, advertised as Nickparkweb, reminded us their profession is broken. Compliance comes when it has a marketplace of fine details and anonymity — sites where "site security" was born and have launched in a bang. At first, at least through the first few days, they check a torrent; all have starting to be accessed, and can then lose their browsing touch at the next check.

"Our thread is where we broke," says the 57-year-old. "One of the things I remember in the dark was after the spam, because in the first three months from there the person had not heard about it at all and I was constantly helpless as my wife left life."

Encounters of the woman and nature

It's not like the 55-year-old is sobering over Nickparkweb until, however, many people launch to Craigslist now that stock illegal medicines.Lamb's older Greg is a dancer in her basement and a weight and laning player at the Nickparkweb and enjoys aioli. He probably buys some of the illegal medicine here today. The women's private woman is ours and her employer's exception, at least partially, of the law. But her husband is still young and the website might be bad yet. She's able to respond quickly via email in a week, a nationwide spam virus notification system holding back a week or two a week or so while her mother goes out for house repairs for communitywork, utilization etc. In an absolute heartbeat, she's meeting with her husband today for dinner or other occasion.

"Working to something that ultimately matters is only the first day," she says. "When

Figure 7: SEDD-Uniform Small. Unconditional

carried out 171 parliamentary committee rules before it was released by results.

On Sunday, the Indonesian government organised a massive riot. Oh, the loyalist Indonesian Republican Party (PEN) pushed the communist government to take an important minority to Indonesia to show how it would remove measures about their religion from the government and prevent blasphemy.

Reuters publishes details Indonesia's anti-LGBT government allowing the community in to perform on Sundays has claimed it would threaten the safety of the country's judiciary, the Organization for Rights Watch (OSF).

Nonetheless, Indonesia is one of the only countries which places routine legal restrictions against religious minorities, including those deemed secular or a religion, who are elected in parliament.

The government prohibits foreign ministries to be run through the huge majority of lawmakers appointed since 2011 most of parliament.

"For LGBT groups, sentencing has become a major topic on the politics. The LGBT groups have continuing to carry out killings and abuses, which seriously disturb the social events of the earth. You see Gaza, of course, to military deaths," said Idelano Gaiyas, a refugee worker and a resident at the Jakarta Proxen Party office. PEN arrests were made in April to counteract a homophobic speech.

He said he helped highlight anti-homosexual extremism and the persecution of the gay community. The Jakarta MP was sacked late last year from his job because of concerns of the number of gay victims in Indonesia and homosexuals.

He said he paid terrorists to severely curtail his community's ability to respond to the threat of civil disobedience and arresting.

"The anti-LGBT government's other ways of faceing people in the government range from groups like Hezbollah. One man was killed in 2009. Police were trying to investigate smuggling explosives linked to a gay worker, but failed to apprehend a man who joined the 2001 LGBT/gay revolution," a spokesman for the official Indonesian government said.

Islamic groups say rights laws try to compel activists and refugees to ignore the threat of persecution in the courts.

"It's really hard to escape from sections of Indonesia's opposition to expect speedy trials," Mantas said.

In the courts, Indonesian governments try to combat discrimination. Among the central reasons for trials is to collect on and hear challenges of cases about harassment and overt discrimination.

Criticised the speeches during would-be hearings produce evidence to talk to the police or assure conviction of the perpetrators of the crimes.

They are also often used as an outlet for classified information, to keep investigators from interviewing victims thickly.

"It's like the legal system," said. "There's such a complex system on it, that seeing what has happened in the past really is difficult."

That same court will be investigating the case of S.6 and electing witnesses to testify in consultation with terrorists during parliamentary proceedings in a public trial.<|endoftext|>Som is when it makes sense that June — not only only the strongest ever June at 17 but, after the previous 10, the third-fastest June since 1974 — is built to a sixth consecutive month.

That would be a prediction for many of the "Miami Hispanics," and to which prices would seem to rise. That number — a decline from about 2 percent to just 12 percent — remains key figures for the so-called winter ahead in which fewer homes are below 80 percent compared with a year ago, said Richard Model, a former county judge and investment adviser at App City and Community Development Bank who took a survey of August, 2017 and the spring. Find home prices from sellout through the end of July.

Model also picked up on a particularly stunning fact: In April and May, during the worst winter, Florida saw a one-year house price increase since 1997 last summer.

Figure 8: SEDD-Absorbing Small. Unconditional

' 2011 moral panic on socio-economic injustice, writes Adam Liberman: Why equal warning gradations are valid studies in moral panic. In popular culture, free speech advocates seem less paranoid than Lou Grivelli, though they should not rule out the possibility that they are being hysterical, since their total fright about a little anarchy – further disastrous if not achieved – are often right. Free-wheeling, hyperpatriarchal social engineering textbooks have tended toward 'autonomy' and gun-toting children becoming sociable teenagers. But if we are ultimately to get over our fear of free-riding pedant thinkers, better should we avoid mass mobilisation over jargon and big grammar vulgarity; and If the Texas revolution we fought for this weekend promises to buck the hell out of obsessives whose incontrovertible Enlightenment response to liberalism has hard ears, why shouldn't we not refuse to cede it – as a matter of principle, there is a disposition after all – to an outworn, nested impatience with ever reverting to deferred pleasures of disinterested action that is sometimes exemplified by Frodo whose sanguinary love of philosophy brings him to the Promise Land?

In classic American university rhetoric, 'experimentation' is equated with blind faith in theoretical truth. It makes a mockery of randomized testing; easier experimentation will simply show you that scientific theories informed by general systems of analysis are equally statistically accurate. Among best novelist voices since the dawn of athleticism were those of Jacques Vallee (first, The Politics of Excuses? ; second, but if history is any guide, most midwesterners will tell you again) and Volker Schlick, who clarified postburial apologetics by which self-knowledge and contemplation are corrected by self experience and solid evidence. For us to have been properly cognizant that disruption of conventional arrangements and institutions such as the church, government, media, economic system, police force and social order bewildered even our naive sense of neoclassicism, democracy, legolito bourgeois hard-luck theories and the direct breeds of sociopathic "random geniuses" would only have become a rotting burden with stressful inertia over the course of centuries, and make it difficult to legitimise Boogie Dees demands for ultimate ruling memos. Their anxiety to safeguard stone-cold goodness against interminable Orwellian ones is probably hindering this progress easily.

Like nothing before, honesty must chasten us from our adherence to an awkward ideal or goal that never really achieved it. 'In on the ground' principles are frequently misused, whether via Uber, a higher-order reality of quantified impulse or the No Mass Paralysis movement, but the most shamefully universal example is gridlock – ticking wheels of gridlock embedded in so many vital consultations in society that the opportunity for deepening conversation over avicingly non-destructive desires may become lost. Hence left-of-center radio comedians, 'lola' advocates and even George Clooney today sometimes dedicate their shows to discerning right-of-center stimulus pilots and ways to strengthen them on pieces of non-boiling petrol. Toward a more forward-looking understanding of our founding myths, straight talk in this field would include addressing defenders of biblically from the South as the mothers of Alphonse, Kipling and Whitaker, attack Finnegans Wake and 'honest citizen' Tony Dawson with a notion of parsimony maxims defining which chicken is pork belly, corruption isn't Booby, killing (in England, women) for no reason, sponsor legal student-burning hijinks and how to prevent 'In-Work trope-making and gaffes'. Unfortunately, global elites and heady resources provide basically the same ambivalence 'Can we really afford economic muckraking? Everything just becomes wrong' as generally seen—but perhaps misguidedly and unfavourably, in these books.

Sure, on some interesting Kansas, Noah's baby, or even Oh Knees as Bush signed a pre-dominantly Trumpish egocentric declaration, artistic monologues suggest genuine changes have occurred, said which affect social's moral standards and hope depending on (examples usually indefinite) objective within-perspective individual study, jury-rigged make-believe relationships, voyeurism becomes a scam, Marx's creation-values should try to convince us 'that Emma and Sasha gave us this expression', the great adage 'focus actually changes the penalty' omits that hey, lines don't change forever, Raymond Carver's Oscars utterances speak better than Obama 3.0, what John Larsson reports in Axas versa endeared in Oda to Hillary, is never bombed or wobbled but shifted his material backing by engaging narratives rather than satellite lying alarms. And complimentary statements with disparate manifestos still distinguish stimulating literature balance within spaces of power and paternal minimization seem divorced from pushing doomed careers towards damaged hands. Varieties of rewriting/claims on the mound and irrespective ethos help intervene on sorts of probability theory in

Figure 9: GPT-2 Medium Analytic Sampling. Unconditional.

1953, he took one in the planned third Bruin Offensive against Northern Germany at Tustin (West Point). Three months he had sent the commanders out to Saracen and the difficulty encountered was tracking down and destroying the submarines there. The Italian submarine hit his mark, but when several hundred thousand had fallen, and against the Germans which had arrived in His city of Sicily, to which he tried to locate a small camp. His second successful mission took place in Bari. The route ran through New York to Madrid and between Mexico, and Morocco. On the day of March the 14th, the suspicious death of British Captain William Warren (B Squadron) on March 20, 1923, opened the way for his second life. Although his two anti-terrorism careers consisted of constant working with Roy Greenspan at the time of World War I for the IMF. Let him call him "The Cardinal." Harriet and I got an opportunity to speak to him, though she told us there was only one name to four others. He was the father of Percy Billings. As in his first case, Warren had been the head of the Australian Air Force. Gates had left once he was accused of sabotage, but he returned de Grin was exiled from power for months. After World War I, he went to Britain as a leader of a group of eighteen members wearing uniforms of the Knights Templar, before going to Italy if needed, helping Sebastiano Riccardo in behalf of the government; by 1916 he was nearly killed in exile by Italian authorities. One of those men, Dr. Sarker, a scientific adviser to the American government, was recognized for his contributions in the English Civil War during the First Kill. In 1914, some say, he had a secret meeting with Hoover on the first day off when the gold standard was signed in World War I. Sarker, we also admit, was a brilliant policeman. Though he was commissioned in December 1921 he was one of only two who did not receive an award, as mass murderer. Some claim, though some dispute, he had gone to the Hague, and he had tried—and even put—on trial the cause of the Hagan Trials. In 1922 while in Buenos Aires, Rose Macdonald, Sarker's divorce solicitor, reported that where her grandmother, Angela Van Ott, lived, she died in Asss, Pennsylvania on March 13. She apparently took her daughter to live with another family. There is documentation of this award in the United States. During lunch as he prepared the report, Harriet pissed his conference father, accusing innocent conflates of agnighting. He told us that he based the previous testimony, in which Alberto C. Rogers and his Captain wereasked to be interviewed, as reasonable. He asked Harriet to explain some of her evidence. We passed on that Rogers himself now said to be the third. He specified, this started, only because he had lost her in the 20 and three years of his case, at her first reading. He extended his invitation to one of the Bow Court's best award winners. It's why he changed. He called on Scott McCain, who appears to have fled from America as the secret source of Elizabeth's evidence. KKR was censored at first sight. In Arkansas he was having made the initial name that had his father's name. He had also mentioned Arthur Zinn's "Gates America" but the name was incorrect. "Then, I had given him Ray, saying I had asked him, 'Is there nothing wrong in this lie? I have discovered nothing?' This was the shot to the head, filled in words from Gates, including: '[T]he Man Nor Wight pilot was consulted with France.' 'No, no, this came out, saying that Arnold Duncan, former Captain of Stowdworth released himself, murdered 14 men at Paris.' I said, 'Then what, then?' He said, 'The Germans want you to send it through America.' And that he would act on it only now, telling me, 'Please you have requested publications for you, especially some of the papers:'. An English Detective was writing me, saying that they had raided his office in Downing Street." Harriet told of the letter that was written in 1893 in which the account proceeded. The letter dated 1900 report from George Hayes, a formerly legendary Army General whose father gave America the results of a destroyed test in the First and Second World War.He quoted some part, "I asked him, 'Have the Germans tried to break everything up?' He said. 'Yes, yes. He will tell you." Harriet testified to the condition of his essay after making a translation that had changed the details of his explanation. "He said that the indications were out on the North Cook. He did not say where the men were odity ITC.

Figure 10: SEDD-Uniform Medium. Unconditional

but a victory will be supported by every president in the past few years and the results will be dramatic. If the Republicans willingly sacrifice their tax revenues to pass legislation that allows them to use messages like "reimpeachment," it will run the risks to be an effective marketing exercise rather than a trasmine.<|endoftext|>In 2012, Jasper Wright, a contractor and former Navy SEAL, made a legal tender offering government and companies "ideal" work on drone development, according to the National Security Agency. But according to the copy below, posted by Copata, Inc. on the FAA website, agencies are prohibited from using a service provided by the firm to receive information or even work on a drone program without security or approval.

For Do, which overall, said it has but never asked the feds to license the remote-controlled software, the Aug. 22 discussions may reveal the long road the agency has taken to protect employees from these rules.

Computer science licensed

Wyman worked as a computer science coach before going to work with the U.S. Secret Service in upstate New York in 2010. Without a license, the Secret Service will have to oversee both the analysts on the software.

"I see this as going to be a matter of choice, but it has been a long road," said Mark McSmith, who specializes in the management of data privacy in the National Security Administration. That includes similar uncertainty about what software must be followed and confidentiality rules under the Espionage Act.

Though the software only takes about four years, he said, for the government to get a license for it, it could take after a federal employee spent a while.

"I think I had to read a lot that nobody was telling the Justice Department about it," he said, adding that "I would guess that it was acquired more recently." But the company lobbied the feds so it could instead oversee its project using a government arm, because of the Bureau of Law.

Do denied the inquiry, and said it made numerous attempts to be in compliance.

"If they've requested to do it and they're still not doing it, don't consider there an artificial interest here," said Flavio Witeli, an agency lawyer, who focuses in cybersecurity law.

To help with Do and Co's troubles, employees find themselves retraining from software products.

Ross Digital, a diverse software and security company in Los Angeles, has the following form on how new leaders can help the agency. "It is a very difficult task" when a person is so over-qualified for a given field, it says.

"I was so surprised that when my team left, they were all high-technology participants, and didn't have any sense why they had joined DARPA," said Ross Digital CEO David Schuner. "They're not over-qualified because the government calls them that."

New employees possible

The quickest way to find out who will rush is a survey, said Jim DeOlson, managing partner at Terra Logging LLC in Hesperia, Calif.

Community companies, neighbors, family or team members, O'Olson said, were essential to how those 40 employees could relocate, but the workload could be huge.

"After a few more days you can start with employees," he said. "If two walked down are sick, then it could cost you jobs around the block as a company."

Riders could also be lured from other companies.

Senior researcher Wagah Herrem of San Francisco drew a proposal that would when taken to the United States: simply building a 100,000-foot (3 mile) cage to look after every DARPA unmanned aerial vehicle at times.

The costs of the office alone would likely range in the hundreds of thousands of dollars, he said.

"I think the work just [being] temporary," the worker said in an interview. "People would go on to say, 'If we could hire X-Z for everything that would go on here, the stress would not lie.'"<|endoftext|>

Figure 11: SEDD-Absorbing Medium. Unconditional

String theory is the fundamental idea that space theory implies a relationship between reality and objects. But what is it really?

That's also the subject of next post. We will discuss several written statements from researchers who have often based our theoretical idea on the Wisenreu-computation principle, where a relationship between reality and objects side no other side. Proclaim that (real or present) an immediate and complete record of our world,they make claims that be said to describe the state at the same of what we can observe. It's a suggestion that we should be working around "dobiverse" frames, and they have nothing to do with the use of monkey consciousness. The moment that will seem like perhaps this is a post of the late '60s. What has distinguished it from these claims? Also, there's a strong feeling that those who are still kicking around the "veil painting" and consensus-author literature have come around advocating a fundamental break from their earlier views.

I don't I should talk here again. Perhaps what we see now is that we contend that the distinction between real bodies and states is inseparable from the theory of these "ological phenomena," and that the relationship between facts and are entangled and not necessarily-existing, because there is perhaps no evidence of connected phenomena at all. While Einstein saw a link between the physicalized properties of the universe and its properties, matter exists and there must be no difference between background particles; just like they are separate objects; when the same properties interact, the different overworld variables expressed as matter are interdependent with this to affect.

The foundation of this argument is to make a similar association to the property theory put forward by Richard Aquinas (1842–1938). In a paper on Perpirus, French biological theorist Richard Field argued that the universe, even in relation to "thing" or the physical world, was not the sole cause or possibility for matter to arise. He was equally pessimistic, as he observed in his paper: "the causes of the creation and rise of a world and heaven were more manifest than matter." So what happens to matter, what happened to land?

(This may go this way: we have "cons" and feel about some things, but we create things — Thomas Aquinas says we can make them so that they create other things. This distinction is the result of having the world mapped out about how we make things up.) But sometimes people may argue that there's a difference between two problems with field theory. In one respect, entities in the universe are not real objects, and in the other it sets nothing in limit to how whatever descended from it is (that) were material, no one little property we associate with matter, including about it. Rather, the world will be material – an example of the properties that it must afford – and specify what it is. The idea is to describe some conceptual framework in terms of what there is about one thing we do have and what is capable of other properties; it would act so that the domain that is built around the very second could be used to justify — in other words.

So the theory treats physics, with an exquisiveness of a general ontological knowledge, in a linear relation to the universe. It is an analogy to special relativity – not a direct analogy to any objects being created. In a remarkable book and probably a manual of metaphysics, Richard Field writes: "the really is about particular relations, as, when something objects interfere with one another, they are dependent on a unique 'material' (whose object or effect he considers this to have different properties on it)." But while the physical property of one necessarily means one is physically real, one is not an object in the physical world, and neither is changing as we know it. So how is that? Theoretically, properties of objects are dependent on some physical object; otherwise physics rules when something in a stationary physical object is something literally physical. This is more of a cogent idea than a modified metaphysics theory that has parallel physical "properties," which re-gates our form of physical entity. Any discussion of the author of thought, which relates to his famous work on incantropy, must be one of four legs. Instead, we have an optimist in a minor theorist status, crippled by a flawed method. What is more productive than few ideas?

At another point in the post and current quote, who proposed a pity for Darwinism observed in his chapter that such theories have little likely influence and mentioned if this theory either practises semantics on the Internet (other than the fad indicated in that wishing it would) or hyperbole (space=hyperbole).

At this time, the entire article has been translated, everything that I draw from it is there's underlying importance. This is research-based

Figure 12: SEDD-Absorbing Small. Conditional in blue.

That is an issue of finding value within the framework of clear market-driven considerations. Some power would have an interesting take on this middle ground, where everybody will look for something. So any new form of the pressure structure embodied in the bylaw market (as well as the brain and life finance) could identify and seize the ostensible challenge of some new technologies, and therefore also solve whether those technologies are genuinely suitable for the possible outcome.

To see issue consistently, a conservative of course would have to reach part of its own conclusion, of which is by consolidating plausible scenarios into a case in itself—that is, scenarios without any political implications at all. Finally, there are political or so many things to do. Parties independent of category go toward course these not places such as actors of organizations are willing to pay for a system that, despite of some aspects of its existence, is an issue for us not them. Ancillary threats are acute in all economic categories and employers are choosing to form them elsewhere. We're asking businesses to engage with organizations to do so and this poster is "New Dancers, a Money for All."<|endoftext|>(with Expositions) http://twitter.com/science/perpework/summons.us/waging-engineer-sur-pent-amount-of-years-771703571

[Interviewer]

*A draft of the 9 August Salon column is on the archived version of Alternet hosted by Ben Sides. They also produce a weekly auto columnist and other blogs.

Post Recommends

Sperrin Baruch, Chair In, Dartmouth

Follow news_opinion

If many people are trying to portray past successes in America's fragile economic recovery as their troubled recovery was in 2015, in retrospect, this is actually just a result of politics. The big plight for Americans in November 2016 is that we were forced to rely upon companies in record closure or a position of being in debt, who would survive the Great Recession by its passage. In so many ways, that's just as far as we get from an uneasy recovery for a historic 8th year of the deepest recession in American history.

While we are often told by elected leaders that conservatives are working to invest in care of Americans, no one seems to doubt that narrative. But for November 2016, this is a significant trend: 2017 is the 4th decade in 65 years. The longest period in 2016 is a the so-called period quieter in its short term with capitalism. In this period 1995, since the Great Recession began, we saw a 4 percent increase in government spending spending over the last 18 years.

These appear to have come about because of the majority of spending cuts made over the 18 months of the recovery found (decades or older). This period has continued into this period. Spending cuts piled up deficits in 2015 and increased our surplus by more than $51 billion in 2015, from $1.3 trillion in 2012. Spending cuts had been expirged in order to sustain our human capital, savings, government health and social programs.

On top are these numbers, it does wonder that analysts are always trying to find just a statistical story or another as people are not looking for anything upward. The economy of America, after the downturn to 2008, will continue to reverse socio-cultural demographic trends from 2015 to 2013. The problem is often trying to determine what remained high with public recovery during this period and where else. Governments have demonstrated a major mechanism for political immigration: stay out, rising, grow in once collected again, and discover population had peaked. Until 2015 there was no private economic recovery during this period as immigrants did during the 2016 fiscal period.

Clearly the change has been associated with economic factors: housing rises and the health effects of life expectancy in the post-2008 crisis – among many trends. Population growth and economic mobility are related to reasons when our country began the Great Recession, and secular tendencies persist. No upward economic trend was produced in the period of 2013, but, may, be related to the fiscal cycle (since 1995) or the increase risk in 2008. This indicates that the current economic crisis will continue unabated for the next 5 years at least.

Figure 13: SEDD-Absorbing Small. Conditional in blue.

"That's a feeling I could give out or leave with a lot of positives out of last season," North Carolina said. "Last season, this felt like the right place later on. It's a pretty solid start the whole way to the NCAA Tournament tournament. I know games will start coming out and I have confidence to go. I know games end up not something out of every game, because of the facilities and some of the players. I have one team that already has not even has their facilities come up. And maybe OK, but only can have the desire to see them into their new stadium this summer. I haven't seen any confirmation that maybe we're going to make a move so I can't give any comment. Nah, I can't."

North Carolina, however, maintains interest in every other aspect of his game than for any other level. He has pointed out how much pain and injury at Duke as it is the average player's experience but insists that it is more simply about his attitude.

"I ever had all of this negative ones during my injury career and that've changed since, and it was a little 'no' in the first couple of February, but there was something positive. As you can tell, that that kept me out for a lot of months," he said. "I just kept going from there. I was all over myself all week, I wasn't even in the process of resting, so I just wanted to play games. I just wasn't so nervous. I just wanted the whole season to recover and see what I can do."

North Carolina will be sure to run off through the first year of he sees what he can get back in line for a tournament appearance.<|endoftext|>I didn't post this discussion last year because I think a lot of climbers have goals for them to be. Speaking of pretty goals, you guess what is in there? Maybe not you. After all, you are. Those athletes are genuinely honest verbally; you. (As a judticist, Attay essentially questioned a set of trike's body forces: post-jumping, dyadicity and dimorphism).

This combination of ego and motivation also isn't beneficial for therapists to athletes to prioritize externalizing their gains in terms of their level of physical placed (research has shown that jumping jacks and abs are insufficient for a healthy profile). Instead, Attay gives consideration to just those reported "basics."

How dangerous does that make an athlete, or just maybe a person

you know you are low capacity

After you attack a mild brain injury supporting an injury, or failure on that last one trade-off, you no longer begin to act in a giggling situation. Without effort, cortisol drains your courage, and you realize you submit to anxiety. It becomes less awkward for someone to log their fitness for you and then lead them back to being active again. It adds a lot to stress.

I have a current personal record of levitating at least 50 repetitions per week in front of a sport I believe and that may only be somebody else is in the works; the type of young female pokesman as well.

I also care to test for each athlete in order of their chances of winning, and I am all about trusting the strength. If you are a pro, consider winning. (Of course, you don't have a record, but I know that picture indicates that you have to climb to climb to win.)

"It tends to be an absolute audition," Attay said, noting that conversation was extreme on one day for one person who he meant to write a report his way up a test-on-and-a-half.

"You want it to come down as close as you can," Infi told Bennett. "But do it twice a day. You'll work hard to apply it, but it will only take up."

Mcm will make sure you are watched

We are seeing now that you need to undergo some critical months of testing that ultimately leads to the end of your health, and that is where you end your chances of doing well. What more often or may not happen is your idea to limit themselves on that risk by weekly assessment those specifically a few weeks.

I know that to make sure you've shown a good level of respect for those administering those tests before:

DI ALWAYS – make sure you are in good shape. As part of this, I will also check to see if you have documented all of your fitness programs or discussions taken during. These put things in context on notes (that's number one) or checklist (mental notes) consists of forgetting old things

Figure 14: SEDD-Absorbing Small. Conditional in blue.

Some popular hiking places include ileceania, Turkey, Greece, and many other foreign countries, such as South American South America, parts of India, East Asia, Russia, North America, China and potential African countries.
– END –
Where are you? It's a easy travel area, so if a hike keeps on going, recommend making sure that you stay aware of your location, and consider this online website 'general maps and reviews.' Currently offering all and best maps for a guide hike on the internet, but you should take care of packing your preferred number of bags and make your trail snacks the "Yes" sort of thing as you're tucked at the back for a long run.
In Poland's remote areas, there's always an okay place to share a bowl of beans with loved ones.
– END –
Having a big house on Olsa.ke, and a long and beautiful mountain, it's very easy to travel to Poland and access your own hiking trails. One of the favorite huts in Poland is Melzazne Kurstrech. To explore the south-western coast and hike the eastern arteries and waterways of Poland. This list is apparently on the company's tourist website.
"We serve all over European industry, the clients are walking, biking, camping, and traveling in the communities - by animal and tuba are riding down Melzazne Kurstrech over hills and aftergones with boats - a ride differentiated by three stylized styles - Loop, Luminous Path, and Wind-Up flat sectioned running as a place where day can shine."
Franklin said it "doesn't matter how far I want to go," he picked up the trails in July, which he dropped to a background later this week.
The Polish authorities, including the Ministry of Polish Tourism, have been working to boost the tourism industry. In the following video from the Polish Ministry publishing a chart on the list of Polish hiking destinations. After counting "Polish locales," this brings in "Slavsans region," "Arsenian West and Hacian Republic", along with "West Calibres and mountains" on it.<|endoftext|>The Coalition of Nurse Aid Delaware is no stranger to the modern world with their training programs. Last summer they posted only about the accredited Delaware program and now I'm thrilled to announce their official website on this post. They are 100% free samples to sign up online for the licensing license program. Participants get the program completely free, as long as they are new:
1) The program requires you to find a facility for the training lessons. This application can help jump forward if you find it.
2) You've got a Delaware license envelope, write your first check. What should you choose on HOA? Become HOA 2017 Now!
Planned Parenthood is a nonprofit organization. It is known for extreme prostitution activity, and sex trafficking, as well as cows, cows, and cows and cows.
S. Del. Code Section 302 – Purient Business
If you don't name yourself "prietary," your business is a thief, or possibly fraud. So, after signing up you for the learning counselor, you may have become concerned that they might do to you things you are not required to do as a mature person or entity under Delaware law, such as mischief, theft, wire fraud,gery, or any form of fraud. Since these companies don't usually have proper permits, they will be found to have just accepted the money in a tax or refund back to the business. Furthermore, in my opinion:
S. Del.C. 304:
60. This Statement, contains:
You and your other licensed business (and that is, no debt related business) carrying out charitable and ethical businesses.
1) You must — by all accounts — have one bank account only.
2) If you any legal object or service that you deem to be charitable, it is carried out first of all. They must pay you first, and it is the employee who pays you – however, that doesn't mean they can claim money as trust just because they thought you needed it.
1. Introduction
When signing up for such classes on that actual website, you need to be kept in school and be familiar with how they are qualified and with different requirements. When you have such consultation, it is a lot more important to keep them informed and that they need your advice.

Figure 15: SEDD-Absorbing Medium. Conditional in blue.

about! I was a nice 'little girl child'. No it wasn't even right now. I had hard backbones. I was light around the skin. A type of me, although I'm more girly. I was in the eyes of both men and women. Gender roles! All those things were a glimpse of where we have a long ways to go. I wasn't in my best. I'm often accused of not caring for myself. Without a doubt, I wasn't in my best at sports. I was lousy at high school as well. The only benefit is that I had being used at every age. It was something I wasn't in my head as much. And it's not just me, it's about me. I saved and care of my family.

I can officially stand up and thank my dad for my appreciation as well, if I wanted to say that much (I feel more every time I think about it). He's really great at it. He put everything between me and my two siblings. He started to feel differently over the years, thanks to when I realized what I wanted to help my sister with cancer. As a biological mother, it seemed like there were several downsides. Plus, it's great, to be happy and be so big, it's wonderful. But at the same time love yourself too, and strive to live life to your fullest. I mean, what are these times? Anywhere I walk, someone asks that question. I want to accept that. Like, "What does this want me to be?"

So I should do this. I should give up. I'm not being stressed out, but constantly stressed out. For the past 10 years, I've actually pumped out more energy than anything else. It's also like it gives me back onto a real quest with my life, it's to be one step ahead of the rest. Same as we get thrown into a fire. The moment you lose your focus, you can reach that goal faster. Knowing my decisions can motivate me, while also having a goal template and letting it help me function can help me do it.

So, I aim for 100,000 steps over the next year or so.

Take pills for weight-exusation medicine, but more cardio, more quality exercise, more caffeine to boost your mood and workout stimulants is good. If you are not more fit or healthy, this is a liporex. Whether you, not only is it incredibly low in fiber but those two things freak you out very thin. Slim you out, how I'm kidding you, I lost when I put you 10 days a day on a wax.

I want you to eat more vegetables, but if you are concerned about health, why the hell don't you be eating micrograms? They don't mean you're fit but make you happy! You are quite terrified of being both nice and thin. I can't decide if there's more here there, but you get my point. The focus on illness and fitness keeps me happy because I sleep and sleep better in times off hot. It's not that complicated anyway. But I suppose not, and I don't think I have to change that!

You know the other healthier things? I was born with incredibly long hair and I just have to admit it sometimes. I care a lot for my hair, I care a lot for them, and other ones, too. I love my skin honestly in Great Sleep, and better than I every-day do. What I keep in mind is lavender. What I shampoo are when in my life. This is carefully, gentle, soft, and regular shampoo; I always run the shampoo a day. I shampoo all the times a week.

It's natural. At least, my hair is hair and it shows. Even so, I shampoo myself all the way up, since it's a pretty direct representation of the world around me. But I still have to shampoo everything.

I carefully enjoy my ears. You know what I will clean them. With the reasons for doing so (to help clean the ears prosperively but avoid earaches). Something natural in life. With constant wash but normal care. This helps to maintain the hair base and repeat clean allows you to put your ear on. bathe three or four a day and seven times a day.

As for shower, I'm not sure. I've always said it was way easier for me to clean. (Although we always make ourselves down) So. I. Did it and I won't do it again. I'm very clean and clean my own shower.

Pare tu Suede?

I absolutely love the feeling of good, good felt and good foot. It's so hard to clean in there. But god forbid I do shampoo in there...and that is why I always shampoo twice a day and shower three times a day.

Figure 16: SEDD-Absorbing Medium. Conditional in blue.

Reasons in Alzheimer's disease
We wrote about these 20 factors and the health benefits of alzetti's disease. For example, a 2013 report in the Journal of Neurotascism, says that the condition is "brain", thereby altering mood and access to limb change. And an updated Case reports that "preliminary reports suggest that a new cure to alzheimer's disease and malaria may have been discovered". People's Week in Music re-published these findings. The 2014 report in the International Journal of Cardiovascular Disease now showed people with dementia had increased risk of death.
Overall, it is quite obvious that disease can lead a person to have fatal problems. Alzheimer's disease has been very well studied. The disease is also not new, and it shows that there are many conditions and risk factors affecting the condition. It is rare that 15 people are born with Alzheimer's disease and few might know who it was. But one study, following lots of older people with the inner symptoms of Alzheimer's, was finding many risk factors.
The protection is evident in a healthy brain, healthy diet, an active lifestyle and less risk for the diseases at home and on the risk for the active lifestyle at work as well as education and other organised lifestyles. The study showed people with dementia were allowed to increase consumption of the amount coffee they drank before they had dementia.
Health-related changes
Alzheimer's disease is by far the main cause of dementia in the US. It is also the main cause of cancer worldwide and second main cause of schizophrenia in the world after TB. That is linked to high levels of inflammatory symptoms similar to those found in Alzheimer's. The same reason young people are more likely to get cancer from tuberculosis and other infections in their lives.
We point to epidemiological studies that follow up thousands of patients plus thousands of studies as evidence that stress is related to the healthy brain and the stressors. And then diabetes occurs most often. What might be the cause? This is why you look at these studies because they can be crucial for a better understanding of the likely pathogenesis.
The robust disease in alzheimer's is closely linked to inflammation. Blood cells are highly susceptible to toxic metals and other things in the blood so they survive the damage of those poisons as well. The proteins from the dead vases in the blood remove their spiny pockets to protect it from damage and doing this do who leave the ulcer to the body. When damaged, the great Alzheimer's disease is devastatingly severe. The brain reacts with strong reactions to the usually weaker proteins causing the inflammatory secretion, suddenly showing a variety of characteristics, including causing archactive rythms in the specific regions that impair the ability to adapt to changes. A study of 60 cases of Alzheimer's disease in the entire

Figure 17: SEDD-Absorbing Medium. Conditional in blue.

