# OpenReview forum: "Discrete Diffusion Language Modeling by Estimating the Ratios of the Data Distribution"
_ICLR.cc/2024/Conference — Submitted to ICLR 2024_

### Official Review · Reviewer_4UNx · 2023-10-17

**Soundness:** 3 good
**Presentation:** 3 good
**Contribution:** 3 good
**Rating:** 8
**Confidence:** 4

**Summary:**

This paper proposes a new loss ("score entropy") for performing score matching in the discrete diffusion setting, building on the continuous-time discrete diffusion framework of Campbell et al. (2022) and the density-ratio-estimation perspective of Concrete Score Matching (Meng et al. 2022). The authors claim that their new objective has a number of advantages over previously-considered losses, and present a number of variants of the objective for computationally-efficient training ~~(although I don't think their justifications are sound, see weakness [W1])~~. They also discuss some alternative forms of sampling algorithms for fast decoding (similar to Campbell et al. (2022)) and discuss how to apply their technique to infilling tasks.

Experimentally, the authors train non-autoregressive generative models using their objective on the OpenWebText dataset, using the GPT-2-small and GPT-2-medium architectures. They present preliminary results of partially-trained models, and show that their model's perplexity bounds are within around 15% of the perplexity of the standard autoregressive GPT-2-small. *Edit: In the current revision they also have compute-matched comparisons to GPT-2-medium, and comparisons to a number of discrete diffusion baselines, with strong results for both.*  They also assess sample quality by computing the GPT-2-large perplexity of generated samples, and find that their samples outperform vanilla autoregressive sampling for a fixed number of network evaluations (or that they match autoregressive modeling with 16x fewer network evaluations).

**Strengths:**

**[S1]** The proposed loss seems intuitively reasonable. The authors motivate it based on limitations of previous score-network-inspired discrete diffusion methods, and argue why their approach should work better. ~~(Although, a lot of this argument seems to rely on unproven theorems; see [W1] below.)~~

**[S2]** The generative perplexity results are quite impressive. It seems that the proposed sampling strategy is Pareto-optimal relative to the fixed GPT-2-small model, when evaluating based on GPT-2-large's perplexity v.s. number of sampling iterations.

**[S3]** The problem of building better non-autoregressive probabilistically-sound generative models is an important one, and the claimed improvements represent an important step in this direction ~~(although due to [W1], [W3], and [W4] below I'm not convinced they've justified their claims sufficiently in this regard)~~

**[S4]** The authors do a good job connecting this work to previous work on diffusion models, and in particular on drawing connections between their score-matching objective and previous work on continuous score-matching diffusion models.

**Weaknesses:**

~~**[W1]**~~ *(addressed in current revision)* The central theoretical claims of this work are incomplete and unsupported, and I am not convinced they are correct. In particular, although much of the paper is devoted to statements about the new "score entropy" and its properties, the proofs are either omitted, incorrect, or only provided in a sketch form.

- Proposition 3.2, which states that their score entropy loss has the right minimum, is never proven.
- Proposition 3.3 and Theorem 3.4, which give alternative forms of the score entropy, have "proofs" that are very handwavey and informal. And I believe these proofs are also incorrect! The derivations ignore the weights $w_{xy}$ and thus end up proving something different than the intended proposition/theorem.
- The central result, Theorem 3.6, is justified only with a sketch which says to apply the (likely incorrect) trick from the "proof" of 3.3 to some unstated result of Campbell et al. (2022). This is nowhere near enough detail to reconstruct an adequate proof.
- Theorem 4.2's proof is also a sketch which does not include enough detail for me to verify its correctness.

Additionally, although the introduction claims that one contribution of the work is a "Langevin corrector framework", this never appears in the paper.

*Edit: The authors have corrected some small errors in their theorems and added detailed proofs for all of them. I believe the theoretical claims are justified now, although I'm not familiar enough with stochastic processes to check everything in detail.*

---

**[W2]** The provided experiments appear to be only preliminary results. For their SEDD-small model, they "emphasize that it is still improving", and for their SEDD-medium model, they state that it "has not started converging". The authors say they will "update our model results as training progresses".

My understanding is that work submitted to ICLR is supposed to be feature-complete at the time of submission. I'm not sure it's appropriate to plan on updating the central results of the submission during the review process.

*Edit: The authors have explained their reasoning below (they meant to pre-emptively reassure reviewers that they could add more comparisons if asked, not to do so unasked). It still seems a bit strange to include comments directed at the reviewers in a paper submission, especially with results that the paper calls "preliminary", since presumably these would always be removed in the final version. On the other hand, if these statements had simply not been added in the first place, I think the initial results would have still supported the main empirical claims, so perhaps this isn't a big deal.*

---

~~**[W3]**~~ *(addressed in current revision)* I found the evaluation criteria to be somewhat imprecise, especially in regards to the authors claims that the demonstrate "for the first time, a non-autoregressive modeling technique that is able to achieve similar perplexity scores as autoregressive modeling".

The authors claim performance is "competitive" with GPT-2-small, but this seems like a subjective statement; the perplexity of their SEDD-small models seems to be a few points higher for everything except the PTB dataset. They also present results for SEDD-medium, a larger model, which outperform the smaller GPT-2 model. However, it's not clear that comparing perplexity across model sizes is fair without controlling for the amount of training compute.

The authors additionally reference the Plaid 1B model from Gulrajani & Hashimoto (2023), which had previously shown strong non-autoregressive performance relative to GPT-2-small (albeit with a larger model and more training compute than GPT-2-medium). That seems to contradict the claim that this work is the "first time" non-autoregressive modeling has been competitive with autoregressive modeling.

I would have hoped for a more rigorous set of experimental results here. For instance, Gulrajani & Hashimoto (2023) give a thorough study of different model scaling law behavior while controlling for training compute; this kind of thing seems necessary to fairly compare with autoregressive methods. (Perhaps much of the performance of the SEDD models here is due to them being overtrained relative to the GPT-2 models.)

*Edit: The authors have added context for their 10% perplexity gap based on existing continuous diffusion results, added comparisions between their medium model and GPT-2 medium, and clarified that their experimental results are not overtrained with respect to GPT-2. The new baselines also provide additional supporting evidence.*

---

~~**[W4]**~~ *(addressed in current revision)* Although motivated as a way to improve upon previously-proposed discrete diffusion approaches, the experiments do not include any discrete diffusion model baselines. Additionally, the perplexity experiments use different evaluation splits and different evaluation methods from previous works, so the numbers cannot be directly compared to previous works. The GPT-2-small comparisons may also be confounded by differences in the dataset or number of training iterations used for GPT-2-small.

It is thus difficult to tell how much of the observed gains are due to the new contributions in this work, rather than being due to the training procedure, base model architecture, or evaluation method.

*Edit: Diffusion baselines have been added, using consistent training, architecture, and evaluation setups.*

---

~~**[W5]**~~ *(addressed in current revision)* The generated samples still seem somewhat incoherent in a qualitative sense. In particular, I found the "infilling" samples in Table 2 to be unimpressive; none of them appear to be meaningful or consistent with the provided prompt tokens.

*Edit: The newer SEDD-medium samples and long-form samples are much more coherent than those in the initial submission.*

**Questions:**

In the appendix, I strongly suggest that the authors fix the proofs, and write them out formally with much more detail than they are currently written.

In the experiments, am I correct in understanding that the "sample quality" experiments are performed on SEDD-small rather than SEDD-medium? That seems important to clarify.

How does the proposed method compare to existing diffusion model baselines like concrete score matching, Plaid, or earlier works? And how does it compare to retraining the autoregressive GPT-2 architecture on the same dataset you are using, with a comparable amount of training compute?

More minor feedback:

- In the intro, what is "this rather straightforward objective"?
- In what sense does the proposed approach induce "an amenable loss landscape"? Do you have any evidence of this?
- In section 2, "Concrete Score Matching" actually learns a vector that is shifted by 1 relative to what you state (e.g. it learns $\frac{p_t(y)}{p_t(x)} - 1$)
- The statement of Theorem 3.4 has a grammar mistake: what is equivalent to the denoising score entropy?
- Definition 3.5 has an incomplete sentence fragment starting "Our parameterized densities ... that satisfy"
- The claim "$Q^{seq}$ is mostly 0" doesn't appear to be explained; I'd suggest adding a note explaining why this is the case.
- In section 3.3 I'd suggest citing the previous works that have studied the "two standard matrices" you propose (e.g. Austin et al (2021) and Campbell et al (2022), and possibly others)
- The equations in 4.1 appear in the middle of the text with no explanation.
- Should equation (21) be conditioned on $x_t$?
- "In principle, we can also bound the likelihoods using Theorem 3.6" appears twice in section 4.2.
- A few citations appear to use `\citep` when `\citet` would be more appropriate (in 5.1 and 6)
- You might want to use a different GPT-2 sample in Figure 2 (b); the current one appears to touch on a sensitive topic.

---

> ### Author Response · Authors · 2023-11-13
> **Rebuttal to Reviewer 4UNx (Part 1/2)**
>
> We would like to thank the reviewer for their extremely thorough review. While the review has expressed positive sentiments about some of our developments, our lack of clarity on the theoretical and empirical front have led the reviewer to be apprehensive about the correctness of our results. The goal of our rebuttal is to smooth this over, showing that our theoretical and empirical improvements are fair and justified. The key points of the rebuttal are:
>
> * **The proofs have been cleaned**. The core theorems for training were correct (in line with our empirical results), and we have updated the proofs to fill in some holes in the proofs. The reviewer should be able to verify them now.
> * **Our comparison with GPT-2 is fair**. The reviewer mentions multiple times that our method may be overtrained when compared with GPT-2. This is not the case (we provide evidence from multiple sources) that our comparison is fair. Additionally, we emphasize that our data setup is very standard (it is more faithful than what it used in previous work like PLAID).
> * **Contextualizing the shortcomings of related work**. The reviewer is reluctant to accept our empirical improvements without including the context of previous work. We emphasize that *all* previous language diffusion modeling work (including PLAID) fall short of GPT-2. We have included an explicit comparison in the main rebuttal to all reviewers.
>
> We hope that the reviewer takes this into account when reconsidering the paper.
>
>
> [W1]
>
> * **On proofs**: We have updated our manuscript to have complete, verifiable proofs. We have double-checked that the proofs are correct.
> The core statements/theorems about score entropy (ie those in section 3) were correct and have been updated to include much more thorough proofs.
> We did find that our statement for Thm 4.2 was slightly incorrect (we had assumed a condition of $\tau$-leaping implicitly). This has been updated to be explicitly mentioned.
> * **On langevin**: We also apologize for the inclusion of the Langevin Corrector bullet point (the content was removed from the paper in a later revision).
>
> [W2]
> * **On reported results**: The work is feature complete (no missing table entries) and thus can be evaluated with the original results. In particular, the submitted results justify our claims (e.g. similar perplexity performance for small models). In particular, authors are allowed to update their manuscript in response to reviewers (and we have done this since other reviewers have asked for medium scale experiments).
>
> [W3]
> * **On the notion of competitive**: the connotation of “competitive” is that the perplexity falls within some reasonable percentage of the baseline (for us, we give this explicitly as +10-15%) under similar architecture sizes. This is consistent with prior work which shows that the variational bound induces a +10% increase in perplexity over exact likelihood methods (Song et al. 2021). While this is definitely fungible, we emphasize that our intention was to draw a stark contrast with previous discrete diffusion methods and continuous diffusion (both of which are at least $2\times$ worse), for which no such claim about competitiveness can be made.
> * **On compute**: we did not overtain our model: our compute is roughly similar to the compute used for GPT-2. In particular, the GPT-2 training details have not been fully released, but various sources put it at 20, 60, or 100 epochs [1, 2], which corresponds to 266k to 1.3M iterations of batch size 512. This is definitely comparable to our 400k iterations of batch size 512. Note that 400k iterations of batch size 512 is a standard amount of compute used in open source re-creations of GPT-2 [3].
> * **On PLAID 1B**: PLAID 1B is around $10\times$ the size of GPT2 small and $3$\times the size of GPT2 medium. It is also trained for 1.2M iterations, which almost surely surpasses the GPT-2 training compute and definitely eclipses our training compute. Since it barely beats GPT2-small and loses pretty handedly to GPT2-medium despite being larger than GPT2-large, we don’t think it’s correct to characterize PLAID as a competitive likelihood-based language model.

---

> ### Author Response · Authors · 2023-11-13
> **Rebuttal to Reviewer 4UNx (Part 2/2)**
>
> [W4]
> * **On diffusion baselines**: we have appended results for standard discrete diffusion methods in the main rebuttal comment and the paper. To summarize, we corroborate existing work which shows that preexisting diffusion methods are poor likelihood learners. We again emphasize that the baseline that we consider is mainly GPT-2, since prior diffusion language models are just not competitive on likelihoods.
> * **On data splits and evaluation methods**: we reevaluate GPT-2 on our data splits, so the results are entirely consistent (they are a fair comparison between SEDD and GPT-2 on the same data splits). Additionally, the method we use for evaluating perplexities is effectively just reusing the CrossEntropy loss function, so it generalizes to all language model methods. This is also a fair comparison since now the evaluation pipeline matches between models (and this also matches the train pipeline).
> * **On training data**: OpenWebText has been found to be an extremely small confounding variable in prior work. The construction of OpenWebText mirrors the original (closed) WebText dataset used for GPT-2 (it pulled the same reddit URLs), and the dataset is used in any other recreation script for GPT-2 [2, 3]. By comparison, PLAID is trained on a cleaned version of OpenWebText2, which has additional data and is cleaned to remove irrelevant data. Since PLAID tests against GPT-2 despite not controlling for this factor, we believe that our OpenWebText training data is more fair than previously reported diffusion model work.
> * **On number of training iterations**: Our comparison is extremely fair here. See above.
> * **On confounding variables**: Our added comparison with the original discrete diffusion method shows that a large proportion of the improvement comes from the score entropy loss. In particular, score entropy results in a $2x$ lower perplexity score when controlling architecture, training hyperparametercs, etc…
>
> [W5]
> * **On generation**: We have given additional text samples in the appendix, which show more coherence and a better usage of the prompt tokens. Note that we are working with small and medium model sizes on OpenWebText (a massive dataset), so a bit of incoherence is expected.
>
> That being said, we emphasize that our work is primarily focused on likelihoods, and our improvements on text generation (which we did not tune for as is already done for autoregressive models like GPT-2 and all other language diffusion models) is a particularly promising (and somewhat unexpected) improvement.
>
> Questions
>
> * **Proofs**: We have fixed up the proofs and are happy to answer any questions about them
> * **Sample quality**: we originally compared SEDD small with GPT-2 small. We have now updated it to compare SEDD small and medium with GPT-2 small and medium.
> * **Baseline diffusion models**: For a comparison against discrete diffusion, CSM, and PLAID, please see the main rebuttal comment or the paper. To summarize: we have corroborated the existing literature and showed that previous methods are not competitive.
> * **Baseline Autoregressive Models**: we have also included a comparison against the training curve of a GPT-2 model. While this has not been trained to completion, the gap in performance is consistent with our final reported gap.
>
> Minor Feedback (in order)
> * The straightforward objective is “learning to generate new samples”, which is much simpler than something like a RL “learn to solve a game” task.
> * The CSM induces a non-amenable loss landscape, which is evident because it does not optimize well.
> * Yes, but this -1 can be absorbed into the network definition, which is what is done in practice.
> * This meant to say “The score entropy loss is…”
> * The phrase continues into the equation definition, ie the probabilities satisfy the following differential equation.
> * Q^seq being mostly 0 follows directly from the fact that transition occur between two sequences that differ by one token. As such, each sequence has d * N neighbors in a space of size N^d, which is incredibly sparse for nontrivial N and d.
> * Sure, citations added.
> * Yes, line 21 does: this has been updated.
> * Updated
> * Different sample added.
>
> [1] https://arxiv.org/pdf/1906.06669.pdf
>
> [2] https://wandb.ai/bkkaggle/lm-finetuning/reports/Pretraining-a-124-M-Parameter-GPT-2-Language-Model--VmlldzoyMjg4NzA#:~:text=Replicating%20GPT%2D2,-I%20tried%20to&text=OpenAI%20trains%20their%20models%20for,epochs%20through%20the%20training%20set).
>
> [3] https://github.com/Dao-AILab/flash-attention/blob/3566596ad867ee415dd3c12616dd50c610176f6c/training/configs/experiment/owt/base.yaml#L28C20-L28C20

---

> ### Comment · Reviewer_4UNx · 2023-11-14
> **Discussion and follow-up questions**
>
> Thank you for the new baseline experimental results and fixed proofs in the main paper, and for the additional clarifications in your response. Most of my concerns with the original submission do seem to be addressed, but I have some follow up comments and questions.
>
> **Comments:**
>
> Regarding [W1]: Thanks for the additional details. I noticed a small change to the statement of Prop 3.3 from the original submission as well, but it does look like the current version is correct. I have a few questions about the proof of Theorem 3.6, described below.
>
> Regarding [W2]: It still seems like planning to add new results to the paper after submission is different from updating a paper to address reviewer feedback, but I'll defer to the AC on this. I agree the main claims of the original submission did not require the "medium" models to have converged, and the additional results do improve the paper.
>
> Regarding [W3]:
> - Notion of competitive: Thanks for clarifying that your target was a ~10% percent gap and putting it in context of prior work, I think the claims you are trying to make are clearer now.
> - Compute: I see, that's important context. It would be useful to add that information to the main paper as well, right now I don't think this is mentioned anywhere.
> - PLAID: I see what you mean. I still think your introduction claim "for the first time, a non-autoregressive modeling technique that is able to achieve similar perplexity scores as autoregressive modeling" is ambiguous in this regard. I take it you specifically want to claim similar perplexity *for a similarly-sized architecture*, and your claim is more about the efficiency of the technique than about the existence of a low-perplexity model? I initially interpreted this claim more broadly, and I'd suggest adding that qualification explicitly.
>
> Regarding [W4]: Thanks for adding the diffusion baselines, which addresses my main concern here. The additional details about OpenWebText and the training iterations for GPT-2 are also informative.
>
> Regarding [W5]: Indeed, I agree the new samples do seem more coherent than the ones in the original submission.
>
>
> **Questions:**
>
> [Q1] Am I correct in understanding that the "GPT-2" star in Figure 3 is a 60k-iterations retraining of the GPT-2 architecture on OpenWebText, not an original GPT-2 checkpoint? If so, I think it would be better to label it "AR Baseline" to match the left-hand figure and avoid confusion. (Assuming I'm right, this seems like good supporting evidence that the effect in Figure 2 is due to the contributions of this work and isn't an artifact of compute or dataset differences.)
>
> [Q2] In Appendix C.2 you say "Since CSM can’t be trained with an ELBO, we simply replaced our score entropy term with the corresponding concrete score matching term". Did you do this during training only or for evaluation as well? It makes sense to do this during training, but I don't think it makes sense to directly compare likelihood-based and non-likelihood-based losses.
>
> For Figure 3 and 4, if the CSM Baseline has a different type of loss entirely, it's misleading to have it on the same y-axis scale. But if you're using Theorem 3.6 to evaluate an ELBO for it despite training with a different loss, that seems fine. In either case it would be useful to clarify this in the paper.
>
> [Q3] In the proof of Theorem 3.6, could you elaborate on how you obtain equations (31) and (32) from the KL divergence term and Dynkin's formula? I consulted Campbell et al. (2022) but I only see one reference to "Dynkin's lemma" there, and its hard for me to see the connection.
>
> Also, there's some confusing notation here:
> - Is $B$ supposed to be $K$?
> - Should $Q$ have a subscript in equation (33)?
> - Should there be a $dt$ at the end of (33)?
> - $Q_t^\theta$ has an overline in Definition 3.5 but not in this proof, are they the same?
> - In (36), should it be $Q(x_t, y)$ instead of just $Q(x_t)$?
>
> **Other suggestions:**
>
> I think it would be a good idea to briefly discuss compute / training iterations in section 5.1. Specifically it seems worth explicitly stating that you train for a similar number of epochs as GPT-2 in addition to matching the datasets and architecture sizes, since this was not obvious from reading the paper but is important for putting the results into context.
>
> Minor:
> - Editing error on page 8: "For all methods, we sample from the true We use the analytic sampling ..."
> - Missing period on page 9: "This is corroborated in Figure 3"
> - Broken math expression for one of the probabilities at the end of page 15

---

> ### Author Response · Authors · 2023-11-14
> **Answering Follow Up Questions**
>
> Thank you for taking the time to review our rebuttal. We are glad that the reviewer's major concerns have largely been addressed.
>
> [W1] Apologies for not mentioning Prop 3.3 changes. This was noticed rather quickly after submission and corrected then (this was a translation error from hand-written notes to latex).
>
> [W2] We understand the reviewers' points here. Our intention was not to update the paper after submission without prompting from reviewers. Rather 1) For SEDD-small, since our method was only trained for 350k < 400k iterations (as is typically done), we wanted to emphasize that we weren't experiencing overfitting (and plucking the 350k iteration as an early-stopping criteria) and 2) For SEDD-medium, since we only train for 150k iterations and can thus only compare against SEDD-small (this was to beat PLAID's previous results showing improvement over GPT2-small), we wanted to preempt any potential criticism that we were unfairly comparing small and medium sizes by assuring that we could compare against GPT-2 medium if asked (note that this exact point was made by Reviewer DQWg).
>
> [W3] We have updated our manuscript to mention these points explicitly.
>
> [Q1] It is a retraining of a GPT-2 autoregressive model. We have changed it to be "AR" to be consistent.
>
> [Q2] We evaluated with ELBOs. This has been made more clear in the appendix and in the main paper figure.
>
> [Q3] We have updated the appendix with this information. In particular, Dynkin's theorem is a generalization of Girsanov's theorem to include Poisson jumps. Similar to what is done in Song et al. 2021 (Maximum Likelihood...), by applying Girsanov's theorem to the KL divergence term between the path measures $\log \frac{dP}{dQ}$, one recovers the term.
>
> [Notation] Fixed. These were typos and the reviewer's fixes were correct.
>
> [Other] We have updated our section 5.1 to explicitly mention this.
>
> [Minor] Thanks for noticing! These have been fixed.

---

> > ### Comment · Reviewer_4UNx · 2023-11-19
> > **Updated review**
> >
> > Thanks for the clarifications and changes. I believe all of my concerns have been sufficiently addressed, and the results remain impressive even with the additional experimental results, so I have increased my score from 3 to 8.
> >
> > One more minor suggestion: I appreciate the clarification for my [Q2] in Appendix C, but it might help to be a bit more explicit, e.g. by adding "Since CSM can’t be trained with an ELBO, we simply replaced our score entropy term with the corresponding concrete score matching term **during training** (although we still report the ELBO for consistency)."

---

> > > ### Author Response · Authors · 2023-11-19
> > > **Thank you for your reconsideration**
> > >
> > > We would like to sincerely thank the reviewer for this detailed discussion and their reconsideration of the paper. We have updated the appendix to make it more clear that we train with CSM but report ELBOs.

---

### Official Review · Reviewer_DQWg · 2023-10-26

**Soundness:** 2 fair
**Presentation:** 2 fair
**Contribution:** 3 good
**Rating:** 6
**Confidence:** 3

**Summary:**

This article is about discrete diffusion models.
Its main contribution is to propose a novel loss function for "concrete score matching" (Meng et al. 2022). This loss function penalizes small values and is hence better adapted to the fact that concrete scores (i.e. distribution ratios) are strictly positive values.
The second contribution is to experiment this novel score loss (with all state-of-the-art architecture proposals) for diffusion-based text generation.
As a third contribution, the authors study extensively this score-matching criterion and its denoising variant and provide an Evidence Lower Bound for likelihood-based training and evaluation.

**Strengths:**

Discrete diffusion models and especially text-diffusion models are difficult but exiting research topics: as mentioned by the authors much work remains to be done before discrete diffusion models can truly rival state-of-the-art autoregressive models on text generation. The main weakness of text-diffusion models is their extremely slow training time when compared to (equivalent) autoregressive models. However their future potential is huge, especially regarding the ability of control they provide.

- I found the paper easy to follow and interesting
- I find the idea of trying a better -- numerically more stable -- score-matching criterion as proposed in this article interesting
- The authors also extend the study of (Meng et al. 2022) and provide an ELBO and a denoising variant of their criterion
- This article may provides a real step toward an improvement of discrete diffusion models

**Weaknesses:**

- There is undoubtedly a lot of work in this article, but I felt that the scientific impact of this contribution is unclear: the main contribution is to propose a new score-matching loss but I see no theoretical evidence and no experiment, be it on a toy example, showing that a simple "quadratic score-matching loss" as in (Meng et al. 2022) would be less efficient than its new "score entropy loss" counterpart.
- The paper lacks of a proper ablation study (be it on small datasets)
- The experiments are only provided on text generation and seem unfinished at submission time (due, I guess, to the huge amount of compute time required to train a medium-size GPT2-like diffusion model)
- On Table 1, the SEDD-medium results are provided, but the equivalent results for medium-size GPT2 must be provided as well otherwise it could be misleading (I hope this will be fixed at the rebuttal time).


Minor remarks:
- typo on page 3 equation 7 : "k\neq i" -> "z \neq x"
- the indices used to write score functions can be confusing to the reader e.g. $s_\theta(x)_y$, $s_\theta(x,t)_j$

**Questions:**

- I found too many ArXiv references: please update your bibliography for per-reviewed versions when possible. For instance A. Campbell et al. 2022 was presented at NeurIPS and should be cited as such.
- On Figure 2 it was unclear to me if the compared models were of SEDD-small or SEDD-medium. I guess it is SEDD-small otherwise it could be misinterpreted as deceiving
- On figure 2: to my experience, using Large-GPT2 perplexity to evaluate smaller models is a good idea but it can be misleading for really bad models (a trivial sequence with a repeated single token being extremely easy to predict, it will reach a very low GPT2 perplexity). I found no solution to this problem. Maybe adding another criterion ?
- Are you planning to provide your code as open source ?

---

> ### Author Response · Authors · 2023-11-13
> **Rebuttal to Reviewer DQWg**
>
> Thank you for your detailed review.
>
> The review seems to appreciate our improvements on prior discrete diffusion modeling works but wants a more detailed comparison to ensure that this is the case. We have tailored our rebuttal with this in mind and hope that our additional work can be factored into your updated review.
>
> Strengths
> * **On training time for our diffusion**: we want to emphasize that we trained our diffusion model using a standard amount of compute. While we did not explicitly compute-match (as GPT-2 training compute is unknown), our compute is similar (see our rebuttal to reviewer 4UNx).
>
>
> Weaknesses
> * **On prior work/a scientific comparison**: apologies , we have added baseline results in the main rebuttal to all reviewers. We emphasize that our comparison was made with GPT-2 because all prior diffusion language models are uncompetitive for perplexities, but we agree that this should have been made explicit in the paper with an experiment.
> * **on CSM**: We have added the CSM baseline in the main rebuttal to all reviewers, where we found that it does not train well (consistent with prior work and our analysis).
> * **on Ablation**: we did not hyperparameter tune any component (the training recipe was created by using prior github repositories for architecture details and used standard hyperparameters such as 3e-4 learning rate, 0.9999 EMA). As such, an ablation of each component is inappropriate. We have ablated the training objective directly (for example, previous discrete diffusion methods underperform compared to our method even with the same architecture).
> * **On experiments**: we have updated the paper (and in the main rebuttal comment) with our finalized numbers. Our updated results show that SEDD-medium vs GPT-2 medium has a similar performance comparison with SEDD-small vs GPT-2 small.
> * **On GPT2-medium in table 1**: we have updated the table to include this. Note that our original table 1 showed SEDD-A Medium just to confirm that it outperformed GPT2-small (which shows improvement over PLAID which required 1B and full compute to beat GPT-2).
>
> Questions
> * **On arxiv references**: apologies, this has been updated. We have added a non-arxiv citation for every paper we could find in our bibliography.
> * **On figure 2 SEDD size**: the model architecture was small. It has now been updated to include both small and medium results (samples are still on small model sizes).
> * **On generative perplexity as a metric**: we are aware of this fundamental issue with generative perplexity. However, as we mention in the paper, this is primarily a problem with alternative (non-analytic) sampling schemes. For instance, greedy sampling often degenerates into a repeated word (e.g. “the the the …”), which hacks our metric (resulting in extremely low perplexities).
>
> &ensp; In general, generative perplexity is a valid metric when comparing ancestral sampling, especially when paired with standard perplexity. The probabilistic interpretation is that the standard perplexity represents the standard KL divergence between the data and the probabilistic model, while generative perplexity represents a reverse KL. If standard perplexity is reasonable (as is in our case), then our probabilistic model “covers” all the modes of the data distribution. If generative perplexity is reasonable (again as is in our case), then our probabilistic model doesn’t place mass on unlikely parts of the data distribution.
>
> &ensp; Furthermore, we manually checked our samples to ensure that our improved results were apparent in the text generation and not a result of some degeneration of the probability distribution. We have included some more text examples in the appendix.
>
> &ensp;  Generally, we found other metrics to be difficult to use. We considered using a generalization of FID, but the issue is that GPT-2 was trained on WebText while our model was trained on OpenWebText. As such, it is hard to produce a consistent validation/test set that would be fair to both models (even though the data is roughly the same, we don’t know the splits).
>
> * **On code**: yes, we are planning on open sourcing the code as well as provide model checkpoints.

---

> > ### Author Response · Authors · 2023-11-19
> > **Almost at the end of the discussion period**
> >
> > We would like to gently remind the reviewer that we are approaching the end of the discussion phase. As there has been a major reconsideration of the paper by other reviewers,  we believe that our rebuttal may have answered any lingering questions posed by this review. We are also happy to respond to additional reviewer questions.

---

> > ### Comment · Reviewer_DQWg · 2023-11-20
> > **Clear improvement**
> >
> > I checked the paper quickly. My major concern was the unfinished SEDD-medium vs GPT-2 medium experiment. I will improve my rating accordingly.

---

> > > ### Author Response · Authors · 2023-11-20
> > > **Thank you for you reconsideration.**
> > >
> > > Dear reviewer,
> > >
> > > Thank you so much for your reconsideration. If there are any lingering doubts in your mind (e.g. CSM, loss ablation, etc...) then please feel free to let us know. Ultimately, we are committed to assuring the reviewer that our paper fully represents a "real step toward an improvement of discrete diffusion models", as mentioned in the initial review. As such, we hope to address any remaining questions.

---

### Official Review · Reviewer_LEE7 · 2023-10-31

**Soundness:** 4 excellent
**Presentation:** 3 good
**Contribution:** 4 excellent
**Rating:** 8
**Confidence:** 4

**Summary:**

This paper demonstrates a new criterion that can be used to train diffusion models for language modeling.  Building on previous work, the author suggest training a network to estimate p(y)/p(x).  The key original contribution of this paper is a training criterion that is non-negative and reflexive, as MSE would be, but that also imposes a constraint requiring the network output to be non-negative.  Essentially, rather than being symmetric about the optimum in score space, the new criterion is symmetric about the optimum in log score space.  The authors argue that the new criterion is theoretically better justified.  Two simplified versions of the proposed criterion are proposed, and from one of them, a score-matching training update is proposed.  The general diffusion Markov transition matrix is argued to be too memory-expensive for practical use, so simplified transition matrices are proposed, one which tends toward a uniform distribution, one which tends to place all the probability mass into the mask label.  Using these simplifications of the proposed criterion, the authors train and demonstrate diffusion-based text generation that is comparable to GPT-2.

**Strengths:**

The proposed criterion is extremely well justified from a theoretical point of view.  The simplified criteria for model scaling are well justified.  The derivations were fun to read and follow.  The experimental results are compelling.

**Weaknesses:**

The only weakness is that, while presenting so much detail about the scaling properties of the proposed criterion, the paper omits to explain the unusually complicated form of the criterion itself.  The derivations give wonderful consequences of Eq. (9), but don't really explain where Eq. (9) comes from!  This might be relevant because I think there might be a small typo in Eq. (9).  I am almost able to derive Eq. (9) by making the assumption that it is a Bregman divergence between s(x,y) and p(y)/p(x), using -log as the convex function, which would totally make sense, because it would guarantee that your score divergence is non-negative, reflexive, and convex in s(x,y); these properties are stated in the paper, but are not proven in the paper, perhaps because they follow naturally from the Bregman divergence.  However, if I derive it in that way, I find one typo in the equation: by that derivation, the last term should not be  (p(y)/p(x))\log(p(y)/p(x)-1), it should be (p(y)/p(x))(log(p(y)/p(x))-1).  Indeed, my correction seems necessary, because log(p(y)/p(x)-1) will often be taking the logarithm of a negative number, which would be avoided if you instead calculated log(p(y)/p(x))-1.  Notably, this last term in Eq. (9) is ignored for most of the rest of the paper, since it does not involve s_\theta(x); it seems to be necessary only for the purpose of shifting the criterion upward so that it is strictly non-negative.

**Questions:**

1. Explain a little about the origin of Eq. (9), and check for a possible typo.

2. "from considering the fully connected graph structure and the MASK token" -- say a little more about this.  It seems that Q_uniform is converging toward a noisy distribution in which all tokens are equally likely, which is not the steady-state distribution of all fully-connected Markov processes, so it's not clear to me that "fully connected" is a sufficient motivation for this model -- but the uniform distribution maximizes entropy, and that does seem like sufficient motivation.  Qabsorb is converging toward a noisy distribution in which the mask token replaces all other tokens?

---

> ### Author Response · Authors · 2023-11-13
> **Rebuttal to Reviewer LEE7**
>
> Thank you for your positive review!
>
> For equation 9, our motivation was initially to scale the gradient of the concrete score matching to make it more amenable for the positivity of the ratios (as we mentioned in the paper). This led to the core $s - \frac{p(y)}{p(x)} \log s$ loss function, from which we found a connection with the ELBO. Ultimately, the constant term was added to ensure that the ELBO behaved properly (greater than or equal to $0$).
>
> However, we reexamined the equation given in the paper. The reviewer is right, there is a typo! The constant should be of the form $a(\log a - 1)$ instead of $a \log(a - 1)$, where $a$ is the ratio. In particular, this was the constant used for denoising score entropy for the ELBO. As such, the reviewer’s points about Bregman divergence are particularly relevant. We have updated the paper to include the Bregman divergence motivation and will credit the reviewer for their astute observations in the acknowledgements (note that the acknowledgements are omitted submission as per ICLR policy). Thanks again for your suggestion here.
>
> For uniform, perhaps it would’ve been more precise to say that the graph structure is uniformly/simplicially connected, with an equal edge weight for all edges. This was chosen mostly because this graph appears quite a bit in prior work.

---

### Official Review · Reviewer_XAVS · 2023-11-01

**Soundness:** 3 good
**Presentation:** 2 fair
**Contribution:** 2 fair
**Rating:** 5
**Confidence:** 3

**Summary:**

* The authors propose SEDD, a generalization of score-matching to the discrete space, which improves upon existing approaches such as CSM (concrete score-matching).
* SEDD satisfies certain desirable properties, such as consistency, ELBO/likelihood-based training, and scalability to high dimensions.
* The authors demonstrate SEDD in the context of language modeling and train a discrete diffusion model that closely matches the performance of an autoregressive baseline (GPT-2).
* SEDD is also capable of arbitrary infilling and provides an option for tradeoff between quality and speed.

**Strengths:**

* The paper is well-motivated and addresses an important area of research that is of interest to the larger community.
* SEDD generalizes score to the discrete domain and improves upon CSM by addressing its limitations (i.e., infinite KL divergence) and satisfies a number of desirable properties that make it suitable for score matching.
* SEDD models achieve competitive metrics compared to GPT-2 on a variety of standard datasets, which suggests the robustness and generalizability of the method.

**Weaknesses:**

* The experiment lacks good baselines. Although the paper claims to improve over concrete score matching, they do not consider CSM in their baseline and only compare the proposed SEDD with an autoregressive model (GPT-2 small). Moreover, SEDD-medium is compared with GPT-2 small.
* The experiment appears inconclusive or incomplete. The model is still being trained, and the authors claim that it has not converged yet; the experiment on the 1 billion-word dataset is said to have encountered unexpected errors, without elaboration.
* Certain design choices lack justification. The SEDD model uses rotary embeddings instead of learned positional embeddings, as in the GPT-2 baseline. In the absence of ablations, it is unclear how much this decision impacted obtained results.
* Not all variables and notations are clearly specified, making the paper difficult to follow at times.

**Questions:**

* Could you add a CSM baseline for an accurate and fair comparison between SEDD and CSM and also provide final, updated metrics of converged models for clarity of analysis?
* How does SEDD-medium compare to GPT-2 medium?
* Is SEDD comparable with GPT-2 in terms of latency in inference? Although they may have a similar number of network evaluations (as in Fig. 2), AR models can leverage techniques like KV-caching, which probably makes them significantly faster than diffusion models.
* What does z denote in Eq. (7), and what does Q_t (y, x) mean in Eq. (6) (Q_t is originally introduced as a matrix in Eq. (5))?

---

> ### Author Response · Authors · 2023-11-13
> **Rebuttal to Reviewer XAVS**
>
> Thank you for your review. The review seems to appreciate our comparison with GPT-2 but has asked for more comparison with baselines and a more thorough analysis. We have tailored our rebuttal to address these points and hope that the reviewer can factor this into their review.
>
> To address the weaknesses
> * **On baselines**: We have updated our results to include other diffusion model baselines (see main rebuttal to all reviewers), which shows our improvements over the baseline diffusion methods.
> * **On Concrete Score Matching**: Please check out our overall comment to all reviewers. As expected, CSM results in a much worse optimization because of the issues we describe.
> * **On experiments**: We have updated our finalized numbers in the comment to all reviewers (and in the updated manuscript). This includes the medium size results, which show that SEDD-medium vs. GPT-2 medium has the same performance comparison as SEDD-small vs GPT-2 small (similar perplexities, better generation quality). Note that our original submission only compared a preliminary version of SEDD-medium with GPT2-small to show that our method scales (and also to beat the previous PLAID method which required 1B parameters to beat GPT-2 small).
> * **On 1BW**: For the 1BW dataset, we found that the public implementation of GPT-2 and 1BW on huggingface resulted in abnormal behavior (unable to reproduce reported results). As such, we include it for completion purposes but use the reported results.
> * **On design choices**: We did not hyperparameter tune our models but rather used what was most convenient (ie existing in previously used codebases). For rotary embeddings, this was used in (Gulrajani & Hashimoto, 2023) and is the basic interface with flash attention. However, our comparison with other methods used our model as well; the performance improvements seem agnostic to model choice but are rather tied with the training objective.
> * **On variables**: apologies. We have updated our manuscript which should help with this problem. If you have a more specific equation that is unclear, please feel free to let us know.
>
> To answer questions
> * **On CSM + baselines**: We have done so. See above. In particular, CSM behaves poorly, as expected.
> * **On medium sized models**: we see a similar dynamic between SEDD-medium and GPT-medium as we do in the small case. In particular, we get a similar performance with perplexities and see the same generation improvements. This has been updated in the main figure.
> * **On latency**: KV-caching does greatly increase the speed of sampling for AR models, as we allude to in our limitations section. For our purposes, autoregressive generation with KV cache is normally about as fast as 64 step diffusion, but we emphasize that a valid latency comparison would require comparison over, e.g. multiple GPU types, optimized pytorch operations, etc… which is outside the scope of our paper. Furthermore, we leave the question of improving our non-AR generation (whether through theory, algorithmic tricks, or GPU optimization) to future work.
> * **On confusion in equations**: (7) the summation should be z != x, which explains what z is (as we iterate over that index instead of k, which was a typo). (6): Q_t(y, x) refers to indexing at the y-th row and x-th column (this is done with parenthesis to avoid overloading the subscript).

---

> > ### Author Response · Authors · 2023-11-19
> > **Approaching end of discussion phase**
> >
> > We would like to gently remind the reviewer that we are approaching the end of the discussion phase. As there has been a major reconsideration of the paper by other reviewers,  we believe that our rebuttal may have answered any lingering questions posed by this review. We are also happy to respond to additional reviewer questions.

---

### Official Review · Reviewer_dGxc · 2023-11-06

**Soundness:** 3 good
**Presentation:** 3 good
**Contribution:** 3 good
**Rating:** 6
**Confidence:** 4

**Summary:**

This paper introduces a novel training objective called "score entropy" for discrete diffusion models, which is analogous to score matching used in continuous diffusion models. The proposed discrete diffusion model achieves comparable perplexity scores to autoregressive models. It can also generate higher quality samples with fewer function evaluations compared to autoregressive sampling.
The contributions include: (1) score entropy training loss, (2) comparable GPT-2 performance to show the potential of diffusion models.

**Strengths:**

- Thorough theoretical analysis about the diffusion weighted denoising score entropy.
- Better generation quality than same-scaled GPT-2

**Weaknesses:**

- Evaluation is a little bit weak. Like, lack of comparision between previous discrete or continous diffusion mdoels, including the sampling speed and generation quality. No specific numbers of  sampling speed (only the caption of Fig 2 mentioned once). No quantitative evaluation for infilling tasks, just showing some examples.
- Some motivations are not clear. Section 4 is not well presented. Why the design of this denoising scheme is needed? If it is designed for speedup, you need to explain two things: (1) no detailed experiments or ablation study about this strategy (2) some discrete diffusion models can sampling within several steps (like~10), and in such condition, discrete diffusion models already have the advantages over generation speed, so what's the difference between theirs and yours?
- Writing: cictation format (citep and citet) is mixed up.

**Questions:**

1. Can we directly compare the perplexity of AR and diffusion models? (In table 1) The definition of perplexity in AR is a little bit different from the NLL in diffusion models.
2. In Fig 2(a), what is the number of network evaluations? It seems that it does not refer to sampling iterations. GPT-2 is also in this Figure, what is the number of network evaluations or sampling iterations of GPT-2?
3. It seems that you use $Q_t(i,j)$ (Eq.14) to replace the $w_{xy}$ in Eq.11. However, due to memory limitation, you choose standard matrices Q_uniform and Q_mask, where the value in these matrices is either 0 or 1. Can we assume that the Eq.14 is degenerated and we actually do not have the soft weighted denoising score entropy?

---

> ### Author Response · Authors · 2023-11-13
> **Rebuttal to Reviewer dGxc**
>
> Thank you for taking the time to review our paper.
>
> The review seems to want a better contextualization with previous work, especially for generation quality. We emphasize that our work was primarily focused on perplexity (outperforming previous diffusion model work for this metric and finally challenging autoregressive models), as perplexity is a critical metric for language modeling. However, we have also presented new evidence (in the main rebuttal to all reviewers) that our generation quality is significantly better than that of prior diffusion methods (despite our method not being trained for sample generation), which should alleviate the reviewer’s concerns. We hope the reviewer can take these points into consideration when reconsidering their review and rating and are happy to answer any more unanswered questions.
>
> To address the weaknesses
> * **On weak evaluation**: Our evaluation is complete as it is a thorough comparison with the baseline (which is GPT-2). We evaluated both likelihoods (in the form of perplexities) as well as generation quality (using generative perplexity), which is typical for any generative modeling task. Note that the previous paper which has scaled language diffusion models to GPT-2 experiments (PLAID) does not test generation quality, so our evaluation is more complete than what has been done in the literature.
> * **On infilling qualitative examples**: The use of qualitative examples for infilling is typical for diffusion modeling (e.g. the inpainting experiments in Song et. al 2021, conditional sample generation for PLAID), and this paradigm is especially applicable in our case since GPT-2 does not have this functionality as an autoregressive model.
> * **On lack of comparison with other diffusion methods**: we have added experiments showing that our method achieves better perplexities and generation quality when compared with previous methods.
> * **On comparing sample generation vs other diffusions**:  see point above.
> * **Specific numbers for sampling speed**: the number of network evaluations/sampling iterations was chosen from [64, 128, 256, 512, 1024, 2048].
> * **On denoising scheme:** This denoiser was motivated by tweedie’s theorem for score functions in continuous space. It results in a slightly improved performance (Appendix C.2) and is included to show how the information given by the concrete score can be used to augment the sampling process rather than just using the Euler step.
> * **On difference with previous discrete diffusion model sampling schemes**: we are fundamentally learning a different value (the concrete score, not the predicted mean). As such, our developed samplers are fundamentally different since the information we are given is different from previous models.
> * **On fast discrete diffusion model sampling schemes**: we are not aware of any prior discrete diffusion model that is able to generate high quality text (e.g. OpenWebText or even 1BW training data, GPT-2 scale models, thorough comparison with baseline autoregressive model) with 10 steps. We are aware of some papers (such as Zheng et al.) which show reasonable generation quality on much simpler tasks, but this has not been shown to scale and evaluates using the statistically weaker GLUE score. If you have a reference in mind, please feel free to let us know and we will take a look and continue this discussion.
> * **On citep vs citet**. Apologies. This has been fixed.
>
> To answer questions
> * **On NLL vs perplexity** Yes, we can directly compare NLL vs Perplexity. This is done before in the previous diffusion language model papers and follows exactly from the fact that $e^x$ is monotonic:
>
> $$ \mathrm{NLL bound} \ge \mathrm{NLL} \implies e^{\mathrm{NLL bound} / L} \ge e^{\mathrm{NLL} / L} = \mathrm{Perplexity}$$
>
> &emsp; We emphasize that our results are entirely consistent with the existing literature. As such, our results indicate that we have effectively closed the perplexity gap (with autoregressive models) that plagued prior diffusion work. This is again a very important point, since language modeling experiments are most often tested on perplexities. Furthermore, since we can only report a bound, our true perplexity is likely much closer with GPT-2 than the given +10% difference
>
> * The number of network evaluations refers to sampling steps. GPT-2 uses 1024 since it is a fixed length autoregressive generation. We vary our number of sampling steps from 64 to 2048 as previously mentioned.
> * On “degeneration of equation 14”. The matrix entries will change based on the noise schedule $\sigma(t)$ and are not $0$ or $1$. Even in the absorbing case, when many values are $0$, we can still recover the relevant scores used for parameterizing the reverse process.

---

> > ### Comment · Reviewer_dGxc · 2023-11-14
> > **Thank you for clarification**
> >
> > Thank you for your clarification! Some further questions:
> >
> > 1. In your updated Fig 3, which discrete diffusion model in specific do you use to compare?
> > 2. I think Zheng et al 's RDM model can do translation task in ~10 steps sampling while maintaining comparable generation quality with Transformer AR model, which is not a simple task. Can you further compare your model with RDM about the sampling speed? In your model, if you use 2048 steps (iterations), it still requires many sampling steps, so I didn't get how to speed up.
> > 3. It is weird to regard GPT-2 as1024 fixed length autoregressive generation. Because in real application scenarios, GPT model will generate [EOS] token to stop the generation. So I think the comparison is not applicable.
> > 4. When you saying *learning a different value (the concrete score)*, can you further explain what does the concrete score refer to specifically?

---

> ### Author Response · Authors · 2023-11-14
> **Answering Further Questions**
>
> Happy to answer questions!
>
> 1. The baseline is a version of D3PM. This is mentioned in Section 6 and in Appendix C.2.
>
> 2. We have reviewed the RDM paper (including the most recent submission to ICLR). There are several points that we need to make:
>
> * It is the general consensus that machine translation is a much simpler task than language modeling, especially at the scale of GPT-2. RDM's improvements need to be contextualized with that setup (especially as RDM does not compare on any language modeling task, which is what discrete diffusion models were originally built for).
> * (Related to the above). In RDM, the non-autoregressive CMLM baseline from 2019 is very competitive (16 iterations, very close GLUE scores). This indicates that the task is rather approachable for non-autoregressive language models (indicating that the task is rather simple). Meanwhile, we don't believe that non-autoregressive models have been competitive on large-scale language modeling tasks, which means that these tasks are significantly more challenging.
> * Although RDM can do machine translation in 10 iterations, it is unclear how many sampling iterations were given with the autoregressive baseline. Note that this is never reported in the RDM paper, but these machine translation tasks typically have significantly shorter lengths (looking at one dataset [1], it seems that the sequence length could easily be <20). If this is the case, then 10 iterations is only ~1/2 the number of iterations of autoregressive models.
> * RDM never outperforms the transformer baseline. We outperform GPT-2 using just 64 steps (1/16 the iterations).
> * We can use less than 2048 steps. If we sample with 64 steps, this will be a speedup over sampling with 2048 steps. As mentioned above, this already outperforms GPT-2.
>
> 3. The comparison is applicable. We are generating length 1024 sequences to compare against (in particular, if we hit an EOS token, we continue generating). We need to generate length 1024 sequences as our evaluation model is expecting this length to compute generative perplexity. This is intended, as the comparison shows that our probabilistic model puts less density on low probability sequences (note that GPT-2 is a probabilistic model for length 1024 sequences). Generally, OpenWebText data is reasonably long (most sequences are several hundred tokens), so generating OpenWebText sequences would take a similar length in practice.
>
> 4. The concrete score is the collection of density ratios $\frac{p_t(y)}{p_t(x)}$. We explicitly talk about this in Sections 2.2, 3.0, 3.1, and 3.2. We also indirectly reference this in Sections 4 (it's in the title and Theorem 4.1).
>
> [1] https://huggingface.co/datasets/bbaaaa/iwslt14-de-en-preprocess

---

> > ### Comment · Reviewer_dGxc · 2023-11-21
> > **Further discussion**
> >
> > My remaining concern is: When you said "We outperform GPT-2 using just 64 steps", you actually evaluate them in generated contents (but not zero-shot PPL). However, in Tab 1, SEDD mostly attains poor perplexities than same-sized GPT-2, and zero-shot PPL is actually a quite important indicator to represent the language modeling capacity of the model. It would be better if you can explain this and show the advantages of SEDD over GPT-2.

---

> ### Author Response · Authors · 2023-11-21
> **Answering Further Discussion**
>
> Certainly! Regarding the reviewer's point: we mention that we outperform GPT-2 using 64 steps for generation quality (the context was that RDM does not outperform using GLUE score, while we outperform using our metric).
>
> For our zero-shot PPL results, we want to emphasize that it's unfair to claim that our results are "poor" (in fact, our paper goes to great lengths to emphasize that these results are highly promising). Our reported results tend to be within a $+10$ \% of the GPT-2 values, and, as noted in the paper, this is a mostly **negligible** gap, especially considering the following points:
>
> * **Our model can only report an upper bound on the perplexity.** In particular, prior work [1] has consistently shown that this upper bound (for continuous space diffusion models) results in a $+10$ \% increase over an exact likelihood (perplexity) evaluation. Our reported bound falls within this $+10$ \% increase, so, assuming a similar phenomena holds, our (non-bound) true zero-shot perplexities are actually competitive with if not surpassing GPT-2.
> * **All previous non-autoregressive language modeling techniques are $> 2\times$ worse for perplexities**. This was shown in prior work, and we have empirically verified that this underperformance also holds in our case. In particular, our results needs to be contextualized with the framing that **no** prior method (that we know of) is competitive with autoregressive modeling for perplexity at this large scale GPT-2 level.
> * **We did not conduct a hyperparameter search** (even though diffusion has a much larger/more unexplored hyperparameter search space). Due to resource constraints, we did not search over any hyperparameters; conversely, GPT-2 hyperparameter searched (including architecture sizes, learning rates, etc...). Furthermore, diffusion models have many more hyperparameters that one can tune (noise schedule, non-causal architecture, loss weighting, etc... in addition to baseline values such as learning rate). Taken together, it is reasonable to assume that, in the future, discrete diffusion methods based off of our SEDD construction can outperform GPT-2 for likelihoods (even with just the variational bound, as was done in Variational Diffusion Models).
>
> Additionally, we would like to note the following about perplexities vs generation quality:
>
> * **Generation quality and likelihood evaluation are somewhat decoupled as metrics.** It was noted in many previous papers (such as [2]) that the likelihood/perplexity of a learned generative model is not enough to indicate if the model is good. In particular, in their example, a model which spits out random noise 99\% of the time does not suffer from a noticeable increase in perplexity, although such a model is obviously unsuitable for any practical purpose (this is an unintuitive outcome from working with high-dimensional probability).
> * **Generation quality as an important metric for comparing different model classes**. (Related to above) it is important to measure the quality of generated samples, especially since our model class is fundamentally different (autoregressive vs diffusion). For example, for images, autoregressive models have long held the SOTA for likelihoods (only recently being beat out by Variational Diffusion Models). However, they have tended to generate very poor image results (extremely unnatural images, bad FID scores, etc...). Meanwhile, the original DDPM diffusion model produced very good generation but struggled on likelihoods. This tends to point to the fact that different model class have different strengths/weaknesses with respect to different metrics, meaning one should evaluate holistically without overemphasizing one metric like perplexity.
>
> As for concrete advantages, we emphasize the following aspects:
>
> * **Better sample quality generation**: as mentioned above, our sample quality generation is significantly better than that of GPT-2, which holds especially true if we use more sampling iterations.
> * **Greater flexibility of generation procedure.** As mentioned in our paper, SEDD allows us to trade off generation quality and compute time. For autoregressive models, no such tradeoff really exists (as tokens must be generated sequentially).
> * **Greater control over the generation procedure.** Also mentioned in the paper, SEDD allows one to control the generation procedure (in this case through inpainting). Generally, diffusion models have demonstrated much more controllability than their autoregressive counterparts and, as our method learns the generalization of the core building block (i.e. the "score function"), many of these techniques should be able to generalize. While this is ultimately outside the scope of the current paper, we want to emphasize that non-autoregressive language modeling intuitively allows one to control the generation of the **whole sequence**, rather than just successive tokens.
>
> [1] https://arxiv.org/abs/2101.09258
>
> [2] https://arxiv.org/abs/1511.01844

---

> > ### Comment · Reviewer_dGxc · 2023-11-22
> > **Thank you**
> >
> > Thank you for your response. I changed the score accordingly. It would be better if you can incorporate the expanded experimental results and details in the final version. The current version appears to have excessive pages on theoretical analysis and fewer pages on experimental analysis. Use Appendix to balance the pages. Overall, it's glab to see the progress on text generation using diffusion model.

---

> > > ### Author Response · Authors · 2023-11-22
> > > **Thank you for your reconsideration**
> > >
> > > Thank you so much for your reconsideration!
> > >
> > > Just one clarifying question. We have already updated the text to include all of the main experiments (unless the reviewer is asking for more text samples). Is the reviewer asking for more experimental analysis in the main text? We believe we can accomplish this despite the ICLR page limit by cutting other sections, but we would just like to confirm.

---

> > > > ### Comment · Reviewer_dGxc · 2023-11-22
> > > > **Respond**
> > > >
> > > > I have reviewed your revised version. However, the experiment section is currently around 2 pages, which I think may unbalance the overall structure of the paper. To address this, I suggest moving more details about the experiment (including ablation) from the Appendix to the main body of the paper. Also, you can consider doing case study about the generated examples, to point out what aspects your diffusion model is good at and what is not.

---

### Author Response · Authors · 2023-11-13
**Overall Response**

# Overall Response

We would like to thank all the reviewers for taking the time to review our paper. Overall, reviewers seemed positive about our formulation and our ability to compete with GPT-2 on several fronts. However, several questions remain unanswered. Below, we’ve compiled some larger-scale rebuttal points. **We urge the reviewers to consult these points and to skim our updated manuscript, as they will likely answer any questions they have about the paper**. Additionally, we welcome additional discussion.


## On Diffusion Baselines

Many reviewers noted that we did not include baselines for other text diffusion models when evaluating perplexities (opting instead to only compare against the autoregressive GPT-2). Our main intent was to compare with GPT-2, and other diffusion modeling works have repeatedly fallen short here (so GPT-2 is the most natural baseline).

However, we understand the reviewers’ apprehension and have updated the paper to include these other baselines (Figure 3 in Section 6). These results are preliminary 60k iterations, but they are indicative of the final perplexities (since the perplexity curves have plateaued). Please note that a full run of 400k iterations is prohibitive since it takes 2 weeks and 15 thousand dollars of GPU credits, but these results are sufficient to corroborate the results of prior work (showing that other diffusion methods are not competitive on likelihoods).

The summary is as follows:

* Previous discrete diffusion models underperform by a factor of around 2x. This performance gap is consistent with the >2x gap between previous discrete diffusion and autoregressive models.
* Continuous diffusion also underperforms, even more so than previous discrete diffusion methods.
* Concrete Score Matching struggles to optimize (staying at an extremely high loss). This is consistent with prior work and our analysis of the pitfalls of CSM.

Additionally, we have evaluated samples from all models at 60k iterations:

* Previous discrete diffusion models generate noticeably worse samples than our method.
* Continuous diffusion models require many steps (e.g. 2048-4096) to get reasonable generation, but this is still worse than our discrete diffusion.
* Concrete score matching does not produce passable samples.

We also discuss more about training in Appendix C.2. We match architecture and training hyperparameters.


## On Updated Medium (+Small) Results


We have updated results for small + medium results. Our perplexities are:

| Method | LAMBADA | WikiText2 | PTB | WikiText103 | 1BW |
| --------- | ------------- | ----------- | ---- | -------------- | ------ |
| GPT2-small | 45.04 | 42.43 | 138.43 | 41.60 | 75.20 |
| SEDD-small Absorb | $\le$52.21 | $\le$44.75 | $\le$130.49 | $\le$43.14 | $\le$80.70 |
| SEDD-small Uniform | $\le$66.94 | $\le$55.88 | $\le$144.88 | $\le$31.39 | $\le$55.72 |
| | |  | |  |  |
| GPT2-medium | 35.66 | 31.80 | 123.14 | 31.39 | 55.72 |
| SEDD-medium Absorb | $\le$44.60 | $\le$34.85 | $\le$93.26 | $\le$32.97 | $\le$67.91 |
| SEDD-medium Uniform | $\le$51.14 | $\le$39.09 | $\le$100.58 | $\le$37.69 | $\le$79.26 |

Additionally, we have updated our generative perplexity results with the medium sized models, which are found in Figure 2a in the updated manuscript. **The same sampling improvements apply for medium sized models**.

---

### Meta-Review · Area_Chair_jwcw · 2023-12-12

**Metareview:**

The paper presents score entropy, a discrete score matching loss that is more stable than other diffusion methods, forms an ELBO for maximum likelihood training, and can be optimized with a denoising variant.

The strengths of the work commonly mentioned by the reviewers include strong theoretical analysis, better results compared with GPT-2, and generally a promising idea for diffusion models in discrete domains.
However, the reviewers mentioned several weaknesses. All reviewers had concerns about lack of other diffusion-based baselines and only considering GPT-2 as the main baseline. Some reviewers noted that the experimental section seemed incomplete at submission time. There were also some concerns about theoretical claims being unsupported or incomplete.

During the rebuttal the authors have addressed some of the concerns including small errors in their theorems and added detailed proofs for all of them. The authors also added preliminary experiments (60k iterations) comparing with some diffusion-based baselines.

However, the experiments seem still incomplete.
In addition, the claim that previous diffusion models fall short of GPT2 might be inaccurate.
For example, SSD-LM (Han et al., 2022), TESS (Karimi Mahabadi et al., 2023), Diffuseq (Gong et al., 2023) report improved results compared with GPT-2, yet there is no discussion or comparison of such methods in related work.
In general, while the paper present a nice idea and the rebuttal resolves some of the concerns, incomplete experimentation at submission time and lack of thorough comparison with relevant baselines are significant weaknesses of the paper.

**Justification For Why Not Higher Score:**

The experiments section were incomplete at submission time. Please see the comments above.

**Justification For Why Not Lower Score:**

N/A

---

### Decision · Program_Chairs · 2024-01-16

Reject